# Microelectromechanical system-based condensation particle counter for real-time monitoring of airborne ultrafine particles

Seong-Jae Yoo[1], Hong-Beom Kwon[1], Ui-Seon Hong[1], Dong-Hyun Kang[2], Sang-Myun Lee[1], Jangseop Han[1], Jungho Hwang[1], Yong-Jun Kim[1]

[1]School of Mechanical Engineering, Yonsei University, Seoul, 03722, Republic of Korea
[2]Micro Nano Fab Center, Korea Institute of Science and Technology, Seoul, 02792, Republic of Korea

*Correspondence to*: Yong-Jun Kim (yjk@yonsei.ac.kr)

**Abstract.** We present a portable, inexpensive, and accurate microelectromechanical system (MEMS)-based condensation particle counter (CPC) for sensitive and precise monitoring of airborne ultrafine particles (UFPs) at a point of interest. A

MEMS-based CPC consists of two main parts: a MEMS-based condensation chip that grows UFPs to micro-sized droplets and a miniature optical particle counter (OPC) that counts single grown droplets with the light scattering method. A conventional conductive cooling–type CPC is miniaturized through MEMS technology and three-dimensional (3D) printing techniques; the essential elements for growing droplets are integrated on a single glass slide. Our system is much more compact (75 mm × 130 mm × 50 mm), lightweight (205 g), and power-efficient (2.7 W) than commercial CPCs. In quantitative experiments, the

results indicated that our system could detect UFPs with a diameter of 12.9 nm by growing them to micro-sized (3.1 µm) droplets. Our system measured the UFP number concentration with high accuracy (mean difference within 4.1 %), and the number concentration range for which our system can count single particles range is 7.99–6850 cm$^{-3}$. Thus, our system has the potential to be used for UFP monitoring in various environments (e.g., air filtration system, high-precision industries utilizing cleanrooms, and indoor/outdoor atmospheres).

## 1 Introduction

Monitoring of airborne ultrafine particles (UFPs, which are smaller than 100 nm), is needed in various fields for human health and yield enhancement in industrial fields (Donaldson et al., 1998; Donovan et al., 1985; Hristozov and Malsch, 2009; Li et al., 2016; Liu et al., 2015). While UFPs have a variety of anthropogenic and natural sources (e.g., soot agglomerates and secondary particles from hazardous gaseous precursors), in urban areas they are largely generated from vehicle exhaust (Kim

et al., 2011; Kittelson, 1998; Shi et al., 1999). Because of dramatic developments in nanotechnology, engineered UFPs for commercial and research purposes have been produced on a large scale. These incidentally and intentionally generated UFPs are more harmful to human health than their larger counterparts. These UFPs have a higher chance of being deposited in the lower respiratory system and are more toxic owing to their larger surface-to-volume ratios, which causes oxidative stress, pulmonary inflammation, and tumor development (Hesterberg et al., 2012; Hext, 1994; Li et al., 2003; Renwick et al., 2004).

Thus, onsite monitoring is needed to assess and minimize UFP exposure. High-precision industries with cleanrooms also require UFP monitoring to trace their sources, minimize the contamination, and thereby increase the production yield. For instance, in the semiconductor industry, where the minimum feature size of the semiconductor devices approaches 7 nm, particles with diameters of a few nanometers are critical (Neisser and Wurm, 2015). Although an air-purification system equipped with an ultra-low particulate filter eliminates the contaminants in the air entering the clean room, it cannot control

the UFPs internally generated during the manufacturing processes (e.g., chemical vapor deposition (CVD), metallization, and wet etching) (Choi et al., 2015; Manodori and Benedetti, 2009). If they are deposited on electrodes of a chip, they cause interruption of the current flow, making the whole chip unusable and thereby reducing the yield of the semiconductor production (Libman et al., 2015). In this regard, ISO 14644-12 has been recently developed to guide how to monitor UFPs in

cleanrooms. To monitor the concentration field of UFPs in these environments where the spatial and temporal variations of UFP concentrations are enormous, portable and low-cost sensors are required to establish simultaneous monitoring at multiple points or dense monitoring networks.

Condensation particle counters (CPCs) are the most widely used UFP detection instruments and are based on the heterogeneous particle condensation technique (Stolzenburg and McMurry, 1991). They grow UFPs to micro-sized droplets through condensation and count them by optical means. Compared to the electrical method (measuring the number concentration of UFPs by electrically charging them and sensing their current), CPCs provide extremely sensitive and precise counting because they are capable of counting individual particles (Kangasluoma et al., 2017; Kangasluoma et al., 2014; McMurry, 2000). Moreover, if a differential mobility analyzer (DMA) is used as particle size selector, CPCs can offer higher particle size resolution than any other particle-sizing instruments (Sioutas, 1999; Stolzenburg et al., 2017). However, commercially available CPCs are bulky and expensive; thus, they are impractical for onsite monitoring where the UFP concentration changes continuously. Although portable CPCs (e.g., model 3007, TSI Inc., USA) are currently on the market, they are still large in size (292 mm × 140 mm × 140 mm) and expensive (~10,000 USD). Therefore, despite their advantages, CPCs are difficult to utilize actively for onsite monitoring applications.

In the past few years, several studies have used microelectromechanical system (MEMS) technology for UFP analysis (Hajjam et al., 2010; Kang et al., 2012; Kim et al., 2018; Kim et al., 2015; Wasisto et al., 2013; Zhang et al., 2016). Because this technology is capable of batch- and micro-size fabrications through the semiconductor manufacturing process, such chips provide cost-efficiency, compactness, and enhanced portability. Most MEMS-based sensors developed for monitoring UFPs were based on the electrical technique. However, these sensors exhibited low sensitivity because their small size means that they have to operate at a small volumetric flow rate (below 1 LPM), and UFPs carry only a small number of elementary charges of their limited surface area.

In this study, we developed a high-performance MEMS-based CPC that is portable, inexpensive, and power-efficient. Our system comprises a MEMS-based condensation chip and miniature optical particle counter (OPC). UFPs are grown to micrometer-sized droplets on the chip and the grown droplets are detected by the miniature OPC. New fabrication techniques including MEMS and 3D printing technologies are applied to CPCs; particle enlargement (i.e., the fundamental process of the CPC) can be realized on a chip-scale system, because the essential elements for growing droplets (i.e., channels, micropillar-type wick, heaters, and temperature sensors) are integrated on a single glass slide. Accordingly, our system is far more compact and cost-efficient than traditional CPCs, even when considering the portable versions.

In addition to its compactness, our system also provides high degrees of accuracy and precision. A quantitative characterization using Ag particles proves that our system is capable of growing UFPs to micrometer-sized droplets, counting them one by one, and thereby measuring UFP number concentration with a high accuracy, which is comparable to commercial OPC. These results show that our system can potentially be used as a portable, low-cost, and high-precision UFP sensor for various fields (e.g., assessing UFP exposure, monitoring workplaces, and tracing particle sources in high-precision industries with cleanrooms). Moreover, when combined with the recently developed miniature DMA, our system should also be able to perform onsite monitoring of the UFP size distribution with high resolution (Liu and Chen, 2016; Qi and Kulkarni, 2016).

## 2 Description of the MEMS-based CPC

Figure 1 shows the operating principle of the proposed MEMS-based CPC, which consists of a reservoir, saturator, condenser, and miniature OPC. To generate supersaturated vapor and hence to grow UFPs to micro-sized droplets, our system utilizes a conductive cooling method, with butanol as the working fluid. The saturator generates saturated vapor by heating the wetted wall with the working fluid. This saturates the UFP-laden sample, which is introduced into the saturator at a flow rate of 0.15 LPM, with working fluid vapor. The saturated sample then enters a condenser, whose temperature (10 °C) is lower than that

of the saturator (40 °C). In the condenser, the hot saturated vapor present in the sample is cooled to reach the supersaturated state. UFPs in the supersaturated vapor act as condensation nuclei and grow to droplets.

The cornerstone of our system is its compact, cost-efficient, and portable features for the onsite monitoring of UFPs. Figure 2a shows our system with the customized circuit. By using the MEMS technology, our system can generate supersaturated vapor and grow droplets on a chip-scale system for significant decreases in the size, weight, and power consumption. The circuit, whose dimension is 90 mm x 65 mm, simultaneously reads the data from the miniature OPC, temperature sensor, and flow sensor (model FS1012-1020-NG, IDT Co., USA), and controls the power of the heaters, cooling modules, and micro pump (model 00H220H024, Nidec Co., JP) via a pulse-width-modulation (PWM) method. In order for our system to be a stand-alone device, the feedback loops based on the proportional-integral-differential (PID) algorithm are implemented in the micro control unit of the circuit, and their gains can easily be controlled using serial communication. The heaters, temperature sensors, and wick are monolithically integrated with the glass slide (Figure 2b). They perform crucial roles: supplying the working fluid to the saturator and condenser via capillary action and generating supersaturated vapor.

As shown in Figure 2c, micropillar arrays serve as a wick of 40 μm diameter, 100 μm length, and 100 μm pitch. The dimensions of the micropillar-type wick were experimentally determined to be capable of pumping the working fluid from the reservoir and spreading it over the entire surface of the saturator to ensure that the saturator wall is always in the wetted condition. SU-8 is a negative-tone photoresist and was chosen as the structural material for the micropillar-type wick because it provides a high patterning resolution (~ 1 μm) and outstanding chemical/thermal stability and guarantees the high durability of our system (Chang and Kim, 2000; Kim et al., 2003).

In order to generate supersaturated vapor with a constant saturation ratio, the temperatures of the saturator (40 °C) and condenser (10 °C) must be controlled to the designed values. For this purpose, the resistive heater was uniformly patterned on the inner wall of the saturator, and a miniature thermoelectric cooling module was attached to the outer wall of the condenser. Both temperatures were monitored with resistive temperature sensors having an accuracy of ± 0.1 °C and located at the outlet of the saturator and center of the condenser, respectively. A customized circuit was used with the PWM method to adjust the power for the heaters and thermoelectric cooler module and thus to control the temperatures.

In the condenser, while the supersaturated vapor grows UFPs to droplets, the working fluid vapor may condense on the wall and clog the channel. Thus, like the saturator, the condenser also had micropillar-type wicks. On a rough surface, the actual contact angle of a working fluid droplet is lower than the contact angle on a smooth surface (Chen et al., 2013). Thus, the micropillar array increases the wettability of the working fluid, suppressing the droplet formation on the wall and draining the condensed working fluid to the reservoir. While the diameter and length were the same, the pitch of the micropillar-type wick (130 μm) was larger than that in the saturator (100 μm).

A spacer, which included the channels and inlet/outlet connectors, was fabricated with a 3D printer in a single printout. The material for the spacer was UV-curable epoxy, which has a high printing resolution (minimum linewidth: 0.3 mm) and can endure temperatures up to 80 °C (Stansbury and Idacavage, 2016). The channel of the saturator was winding to increase the residence time and thus ensure fully saturated vapor. The width, height, and length of the saturator channel were 6, 3, and 150 mm, respectively, and the corresponding residence time at the given flow rate was 1.08 s. The maximum Reynolds number in the channel at the given flow rate was only 32, which means that the sample stream in the channel was in the fully laminar regime.

The detection part of the commercial OPC (Innoair-615D, Innociple Co., KR), which provided a time resolution of 6 s, was used as the optical detector in our system. It consists of the sensing chamber and optics (laser, cylindrical lens, elliptic mirror, photodiode and light trap). The introduced droplets are arranged in a row in the acceleration nozzle (i.e., the outlet of the condensation chip). Then they enter the sensing chamber. The droplets then pass through the place where the condensed thin beam is irradiated. The mirror directs the light scattered from a droplet to the sensing surface of the photodiode. When the laser beam passes through the cylindrical lens, the shape of the laser beam at the focal point is not a point but a very thin

surface. In addition, the acceleration nozzle at the chip outlet is only 0.8 mm in diameter and is located about 1.5 mm below the point through which the beam passes. Therefore, under the condition that there is no coincidence error (the two particles do not pass through the viewing volume at the same time), almost all the grown micro-droplets are counted in the optical detector.

## 3 Fabrication process

As shown in Figure 3, the MEMS-based particle growth system consists of a top plate, bottom plate, and 3D-printed channel. The fabrication process was identical for the two plates. The essential elements on the plates (heaters, resistive temperature sensors, and micropillar-type wick) were fabricated through a simple photolithographic process. An E-beam evaporator was used to deposit a thin metal layer (30/300 nm of Ti/Au) on the plates. Then, a positive-tone photoresist was spin-coated at 3000 rpm for 30 s, and softly baked on a hot plate at 95 ℃ for 1 min. After baking, the photoresist was exposed to UV light (wavelength: 365 nm, exposure dose: 55 mJ cm$^{-2}$) and developed with a photoresist developer. Then, the wet etching process was performed to define the electrodes (Figure 3a). To enhance the repeatability of the temperature sensors, the fabricated electrodes were heat-treated at 300 °C in ambient atmosphere. The SU-8 negative-tone photoresist was used as the material for the micropillar arrays. A 100-µm layer of SU-8 (model 2100, Microchem Corp., USA) was spin-coated onto the plates at 3000 rpm for 30 s and baked at 85 °C for 120 min. Then, it was exposed to UV light (exposure dose: 240 mJ cm$^{-2}$) to define the micropillar array structure. Next, a post-exposure-bake (PEB) was conducted through a two-step ramping process on a hot plate: 65 °C for 5 min and then 95 °C for 120 min. The exposed SU-8 was developed with a SU-8 developer (Figure 3b). The fabricated channel and plates were packaged with polymethyl methacrylate jigs and silicon rubber gaskets (Figure 3c). The cooling modules comprised a thermoelectric cooler, heat sink, and mini fan. They were attached to the outer walls of the condenser (Figure 3d). Finally, the fabricated chip was inserted into a card connector, which was linked to the control circuit board.

## 4 Experimental setup

Figure 4 shows the experimental setup used to characterize the overall performance of our system in terms of three aspects: (a) the clean air supply system, (b) monodisperse particle generating system, and (c) performance comparison system. Compressed air was used as the carrier gas. Any moisture, oil droplets, and particles in the compressed air were removed in the clean air supply system with an oil trap, diffusion dryer, and high-efficiency particulate (HEPA) filter. The purified air was then supplied to the particle generating system at a flow rate that was accurately controlled by a mass flow controller (VIC-D200, MKP Co., KR). Ag particles ranging in size from 3–140 nm were generated by an Ag particle generator (EP-NGS20, EcoPictures Co., KR). The particles were electrically charged by a soft X-ray charger (XRC-05, HCT Co., KR). Then they were classified according to diameter with two types of DMAs: (1) nano DMA (model 3085, TSI Co. Ltd., USA) for particles in the size range from 3 to 10 nm, (2) long DMA (model 3081A, TSI Co. Ltd., USA) for particles in the size range from 5 to 140 nm. Next, the number concentration of the monodisperse Ag particles was controlled (0–24000 cm$^{-3}$) in the dilution bridge system by adjusting the needle valve. Finally, the concentration-controlled and monodisperse Ag particles were introduced into our system and reference instrument, which was either a CPC (model 3772, TSI Inc., USA) or an aerosol electrometer (model 3068B, TSI Inc., USA).

Because of the large difference between the flow rates of the reference instrument and our system, the following procedures were carried out to verify that particles with the same concentration were introduced into the two systems. First, to minimize the particle loss induced from the turbulence at the bifurcation, a flow splitter with a very small angle of split (model 3708, TSI Inc., USA) was used. The tubes which leads to the both systems were electrostatic dissipative to minimize the electrostatic

particle loss, and their lengths were carefully adjusted to match the transportation times. To verify that the particles which were introduced into both systems had the same concentrations, it was confirmed that the counting efficiency was close to 100 % when particles with a size of 100 nm were introduced (it was assumed that they were activated and grew into droplets with 100 % efficiency). Then, while reducing the size of the introduced particles to 40 nm by adjusting the voltages of the DMA, it was confirmed that the counting efficiency remained constant. Through these procedures, it was verified that the concentrations of the particles delivered to the two systems were the same. The loss of our system was characterized using the counting efficiency, since it is defined as the efficiency of the system at detecting the introduced particles, and thereby describes the overall transportation/activation efficiencies.

## 5 Results and discussion

### 5.1 Working fluid transmission and evaporation

In order to characterize the wick capability of the working fluid transmission and evaporation, a rectangular test sample was used. The sample included a heater and temperature sensor, and its length was equal to the saturator maximum length (38 mm) of our system.

To identify the supply capability of the wick, a capillary rise experiment was performed (Figure 5a). The sample was slowly lowered to the surface of the working fluid. Once the bottom of the sample touched the surface (t = 0 s), the rise of the working fluid on the sample was recorded. As shown in Figure 5b, the forefront of the working fluid rose rapidly at the beginning and then slowed down gradually. The forefront of the working fluid reached the endpoint of the saturator in 35 s and thus wetted the whole surface of the sample.

In the saturator, the working fluid transported by the wick is vaporized at elevated temperatures. When the rate of the working fluid evaporation is higher than that of the transportation, a dry-out region is formed. This phenomenon must be minimized because the partial vapor pressure near the wall must be kept close to the saturated condition. Figure 5c shows optical images of the dry-out region formation as the surface temperature increased. The temperature of the sample surface was increased in increments of 10 °C; at each temperature, the system operated for 180 s. The dry-out region clearly did not form when the surface temperature was equal to the designed saturator temperature (40 °C) or even when it reached 70 °C. At 80 °C, the front of the working fluid started to recede, so a dry-out region formed. At 90 °C, the area of the dry-out region accounted for approximately 20 % of the heated surface. These results demonstrate that the amount of the working fluid supplied from the wick was greater than the rate of evaporation in the condenser at the design temperature. Moreover, the saturator temperature can be raised to 70 °C to increase the saturation ratio and hence decrease the Kelvin diameter to enabling the system to detect smaller particles.

In order to obtain optical images of the dry-out region formation, a single glass slide with the patterned electrodes and micropillar array was used. For this reason, there was no flow rate during the experiment. Although, due to the advection, the flow rate affects the area of the dry-out region to some extent, the vapor pressure at which the dry out region started to form (164.1 mmHg at 80 °C) was 8.7 times larger than the designed value of the saturator (18.9 mmHg at 40 °C). Thus, although there was no flow above the butanol surface when measuring the dry-out region, the dry-out region did not occur in the saturator of the MEMS-based CPC under operating conditions.

### 5.2 Droplet size distribution

Figure 6 shows the size distribution of the droplets generated from the MEMS-based particle growth system. Monodisperse Ag particles in the size range from 20 to 140 nm were used as test aerosol and their number concentrations were fixed at around 2000 cm$^{-3}$ by adjusting the valves of the dilution bridge. The sampling time for measuring each droplet distribution was 2 min,

and the corresponding measurement uncertainty based on Poisson statistics was 0.13 %. All the error bars at each data point represent the standard deviations. A commercial OPC (OPC-N2, Alphasense, UK) was used for measuring the droplet size distribution. It has been reported that OPC-N2 was capable of not only measuring particles from 0.4 to 17 μm, but also having moderate counting performance compared to the reference OPC (PAS-1.108, Grimm Technologies) (Sousan et al., 2016). The

5 measurement errors induced from the Mie resonance was not considered in this data. The average droplet diameter ($d_{d,avg}$) was 3.1 μm when particles of 20 nm size, slightly larger than the minimum detectable size (12.9 nm), were introduced. Because the lower detectable size of the optical detector in our system was 0.3 μm, the introduced particles successfully grew into μm-sized droplets that were large enough to be counted by optical means. It was noted that the mean droplet size did not vary significantly above 40 nm. In addition, most of the grown droplets were smaller than 10 μm, indicating that droplets of size

tens of micrometer, which could attach to the inner walls of the particle growth system or optical detector via sedimentation, were barely generated.

**5.3 Size-dependent particle counting efficiency**

The counting efficiency ($\eta_d$) is defined as the efficiency of the system at detecting the particles and describes the overall CPC performance. It is the product of three efficiencies:

$$\eta_d = \eta_{trans} \cdot \eta_{act} \cdot \eta_{OPC}, \tag{1}$$

where $\eta_{trans}$, $\eta_{act}$, and $\eta_{OPC}$ are the efficiencies of a particle passing through our system, growing droplets at the condensation chip, and the OPC at counting droplets passing through its sensing volume, respectively. Because these three efficiencies are strongly dependent on the particle size, in particular for small particles below ca. 30 nm, the counting efficiency must be characterized as a function of the particle size.

The counting efficiency was obtained from the ratio of the concentration measured with our system to the reference number concentration. The reference concentration ($N_{ref}$) was obtained from the electrical current ($I$) measured by the aerosol electrometer (AE):

$$I = N_{ref} \cdot ne \cdot Q, \tag{2}$$

where $n$ is the number of elementary charges (+1) per particle, $e$ is the elementary charge ($1.6 \times 10^{-19}$ C), and $Q$ is the flow rate

drawn into the AE.

Figure 7 shows the size-dependent counting efficiency of the MEMS-based CPC. The size range of the Ag particles was controlled so that the concentration range was 1000-2000 cm$^{-3}$. The sampling times for each data point were 300 s, and the measurement uncertainty based on Poisson statistics was 0.02 %. To evaluate the effect of the temperature difference, the counting efficiency was characterized when the condenser temperature ($T_c$) was 10 °C and the saturator temperatures ($T_s$) were

30 30, 35, and 40 °C. At 40 °C (the design value of the saturator temperature), the same experiments were repeated three times to confirm the measurement reliability. When the saturator temperature was 40 °C, it was found that our system detected 1 % of UFPs with size 5 nm, and the detection efficiency increased sharply above 9 nm. This was primarily because the activation efficiency ($\eta_{act}$) increased when the particle size exceeded the Kelvin diameter (2.34 nm). The transport efficiency ($\eta_{trans}$) also increased, because the diffusivity of a particle decreases with increasing particle size. The counting efficiency data were curve-

35 fitted using

$$\eta_d = \alpha + \frac{(\beta - \alpha)}{1 + (d_p/\gamma)^\delta}, \tag{3}$$

where $\alpha$, $\beta$, $\gamma$, and $\delta$ are fitting constants with values 101.96, 2.00, 12.99, and 4.70, respectively. The corresponding minimum detectable size is defined as the size at which particles are detected with 50 % efficiency. It was found to be 12.9 nm. The detection efficiency was 90 % at 20.1 nm and reached 95 % at 22.9 nm. It was close to 100 % and constant in the size range

from 25 to 140 nm, indicating that the internal particle loss in this size range was negligible.

The minimum detectable size was higher than that of the commercial CPC operating at the same temperature difference. Because the saturator and condenser were close to each other and thereby have thermal interference, the condenser temperature might be higher than its originally designed value due to the heat transfer from the saturator to the condenser. It is expected that this problem can be solved by increasing the thickness of the thermal barrier between the saturator and condenser.

## 5.4 Detectable concentration range

Even if filtered air is introduced into the system, droplets may form in the condenser via homogeneous or ion-induced nucleation. Droplets without UFP nuclei cause false counting, which makes the system read a higher concentration than reality. This phenomenon is critical, especially in low-concentration environments. To evaluate the false counting of our system, it was operated for 1 h with a HEPA filter connected to its inlet. When the temperature difference between the saturator and condenser was set to 30 °C, the average number concentration and counting rate during the measurement period (background concentration) was 0.05 cm$^{-3}$ and 0.125 s$^{-1}$, respectively, indicating that homogeneous nucleation hardly occurred. Thus, the temperature profile was uniformly established inside the condensation chip because homogeneous nucleation typically occurs at low temperatures in regions where the local saturation ratio is high. Owing to the low false count performance, our system can be applied to monitoring UFPs in cleanrooms of class 100 or 1000 whose background concentrations are of the order of 1 cm$^{-3}$ (Liao et al., 2018). In these environments, the measurement uncertainty based on Poisson statistics is expected to be 10 % in a sampling time of 40 s for a given flow rate (0.15 LPM = 2.5 cm$^3$ s$^{-1}$).

We characterized the maximum detectable number concentration of our system by comparing the number concentration with that of the AE (i.e., reference number concentration). Monodisperse Ag particles with a size of 25 nm were used as test particles, and their number concentration was increased at intervals of 3 min. Figure 9 shows (a) the time series of number concentrations measured with the MEMS-based CPC and an AE and (b) a one-to-one comparison of the measured number concentrations by both systems. As shown in Figure 9b, relatively large fluctuations were observed at number concentrations of < 1000 cm$^{-3}$ because of the electronic noise of the AE. However, in the number concentration range of 1000–5000 cm$^{-3}$, the overall difference between the concentrations of our system and the AE was only 4.1 %, which proves the high accuracy of our system. When the concentrations exceeded 5000 cm$^{-3}$, the deviation between the measured and reference number concentrations gradually increased. One of the main reasons was the coincidence error. When multiple particles simultaneously passed through the sensing volume, the miniature OPC could not count them separately. The maximum detectable concentration of our system, which was defined as the number concentration at a difference of 20 %, was 7200 cm$^{-3}$. Thus, when the concentration exceeded 6852 cm$^{-3}$, the logarithmic function was fitted to the response curve of our system, as expressed in Figure 8 b. When the calibration based on the fitted curve was applied to our system, the average difference between both systems was within 2.8 %.

## 5.5 Performance comparison with the reference CPC

The MEMS-based CPC was tested in parallel with a reference CPC. The classification voltage of the DMA was changed to introduce monodisperse Ag particles varying in concentration and size into both systems. The total measurement time was 600 s and the measured data were averaged in intervals of 6 s. Figure 8 shows the measured number concentrations of our system and the reference CPC. When particles larger than the minimum detectable size (12.9 nm) were introduced, our system clearly showed high accuracy and precision comparable to those of the reference CPC: a difference of 4.54 % at a low concentration (7.99 cm$^{-3}$ at 28 nm) and a -9.12 % difference at a high concentration (4544.82 cm$^{-3}$ at 16 nm).

Figure 10 shows the measurement results of our system when it was tilted as shown in the inset. Monodisperse 25 nm Ag particles were introduced and their concentrations were increased in steps from 0 to 4000 cm$^{-3}$. Since the measurement was carried out for about 500 s at each angle, the measurement uncertainty of each section was below 0.01 %. When our system

was oriented perpendicular to the surface, the difference in counting efficiencies between our system and the reference CPC was 2.04%, which was similar to the result of the size-dependent counting efficiency. When a 30 ° angle was applied, the difference in counting efficiency was 7.07 %. At 60 °, the measurement difference compared to the reference CPC exceeded 10 % (16.3 %). Thus, it was found that, at a tilt angle of 60 ° or less, MEMS-based CPC can monitor UFPs without the significant degradation of the accuracy.

The deviation of the counting efficiency induced from applying a tilt angle can be explained by the sedimentation of droplets in the condenser. At 0 °, because the direction of gravity was identical to the direction of the sample flow, the probability that grown droplets hit the condenser wall via sedimentation was negligible. However, with increasing tilt angle, the velocity vector of a droplet perpendicular to the channel increased, which lead to the decrease in the counting efficiency.

## 6 Conclusion

The MEMS-based CPC was developed for sensitive and precise monitoring of UFPs at a particular point of interest. Our system comprises two parts: the MEMS-based condensation chip and a miniature OPC. To achieve compactness, the key elements for growing droplets (i.e., saturator, reservoir, and condenser) were integrated on a 52.5 mm × 60 mm glass slide through a simple photolithographic process and 3D printing. Quantitative experiments with an AE (model 3068B, TSI Inc., USA) and CPC (model 3772, TSI Inc., USA) demonstrated that our system can count UFPs with a size of 12.9 nm and measure the number concentration with high accuracy (within 4.1 % difference compared to AE) in the range of 1000–5000 $cm^3$.

In terms of compactness and cost-efficiency, our system is superior to conventional instruments. The physical volume of our system is only 8.5 % of the volume of the commercially available portable CPC (e.g., model 3007, TSI Inc., USA). Furthermore, its manufacturing cost can be minimized owing to batch fabrication based on MEMS technology. These advantages allow the system to be successfully applied to various fields that require UFP monitoring. Furthermore, if combined with the recently developed miniature DMA, our system can help realize a mini-scanning mobility particle sizer (mini-SMPS) for accurate and precise measurement of the UFP size distribution at particular points of interest.

## Author contributions

S.J.Y., H.B.K., and Y.J.K. conceptualized the study. H.B.K., U.S.H., S.J.Y., D.H.K., and S.M.L. performed the experimental work. S.J.Y. wrote the original draft, and S.J.Y., H.B.K., and D.H.K revised it. J.H. and J.H. provided the resources. All authors discussed and commented on the manuscript and approve of its content.

## Competing interests

The authors declare that they have no conflict of interest.

## Acknowledgements

This work was supported by Samsung Research Funding & Incubation Center of Samsung Electronics under Project Number SRFC-TA1803-05 and by the Technology Innovation Program (10077651, Development of IoT fusion sensor system based on artificial intelligence) funded by the Ministry of Trade, Industry & Energy (MOTIE, Korea).

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

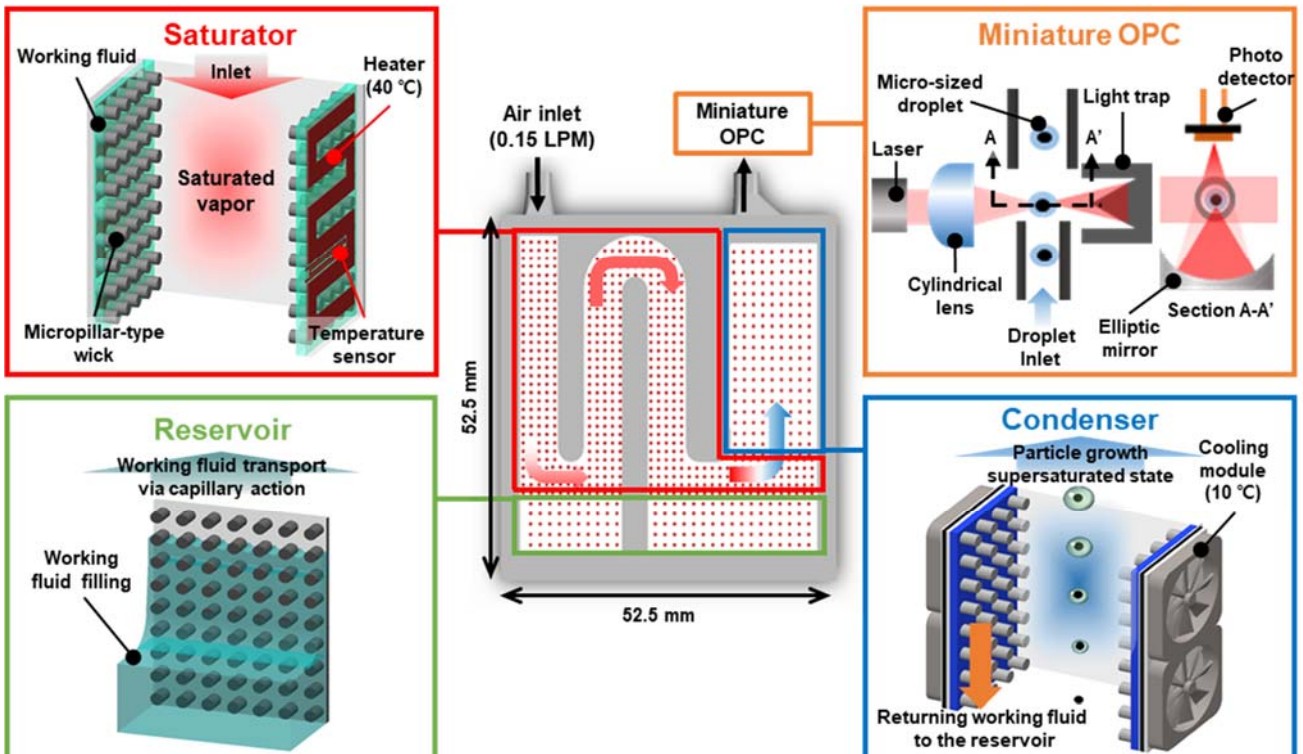

**Figure 1: Schematic illustration of the MEMS-based CPC. Our system consists of four parts: the reservoir, saturator, condenser, and miniature OPC. The reservoir supplies the working fluid to the saturator via capillary action by the micropillar-type wick. The saturator heats the working fluid to generate saturated vapor. The saturated air becomes supersaturated when cooled by the condenser. UFPs grow into micro-sized droplets in the condenser. They are counted by the miniature OPC.**

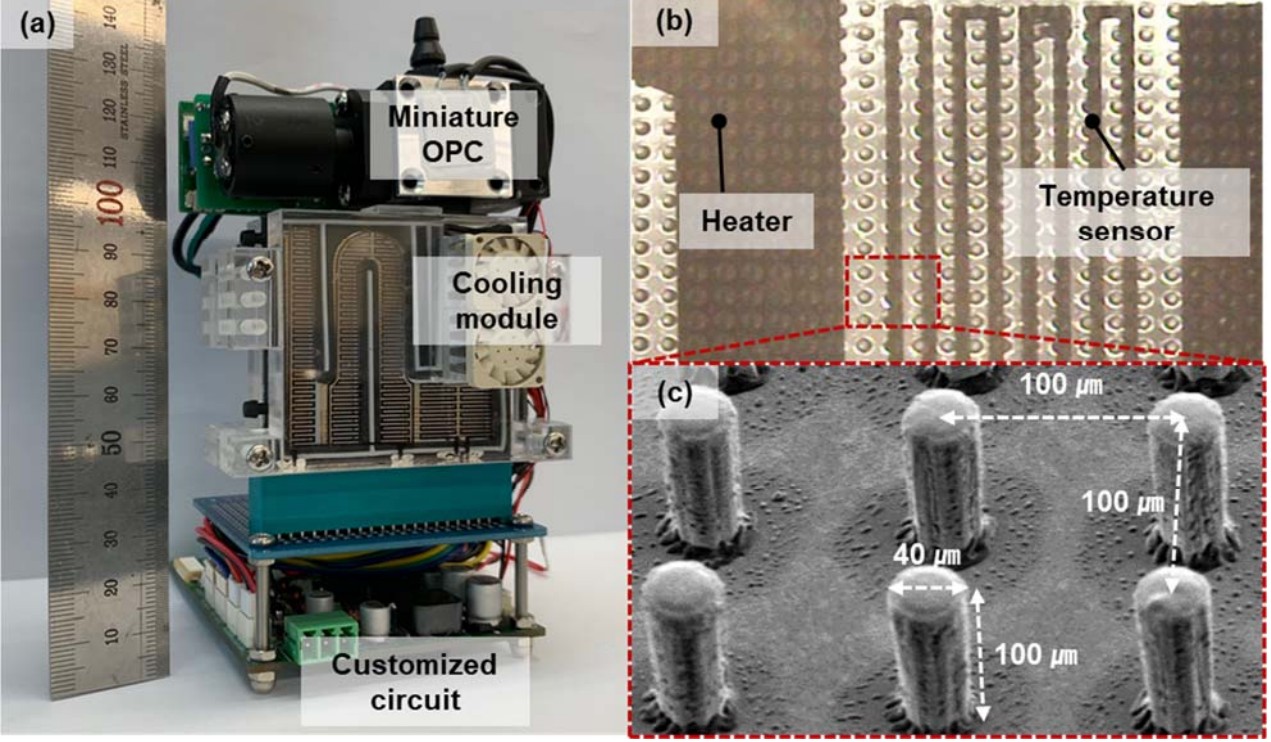

**Figure 2: (a) Optical image of the MEMS-based CPC; (b) magnified image of the heaters, resistive temperature sensors, and wick on the glass slide; and (c) scanning electron microscope (SEM) image of the micropillar-type wick.**

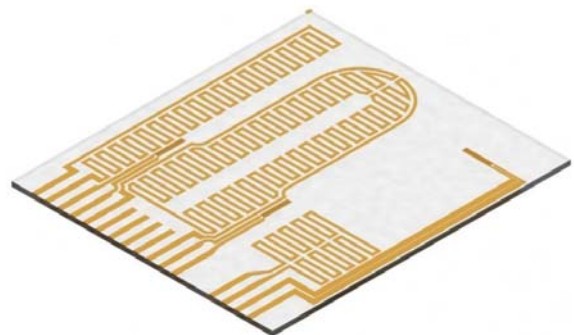

**(a) Ti/Au electrode patterning**

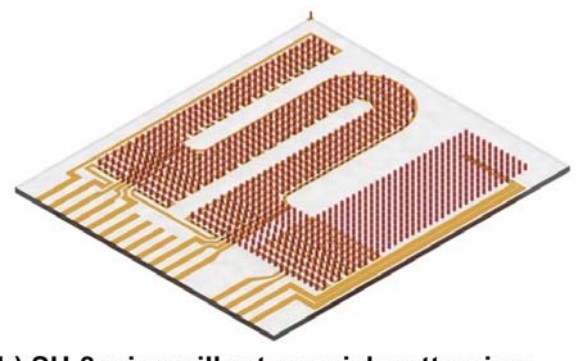

**(b) SU-8 micropillar-type wick patterning**

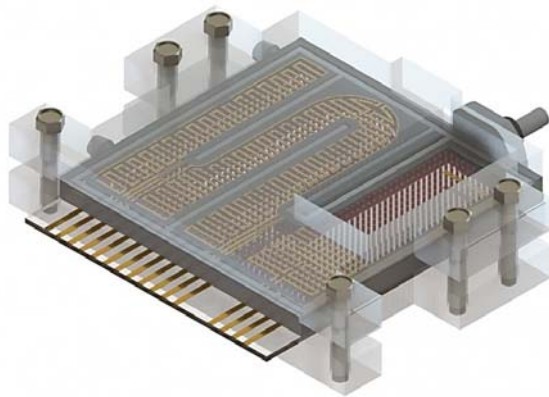

**(c) Packaged using PMMA jigs and silicon rubber gaskets.**

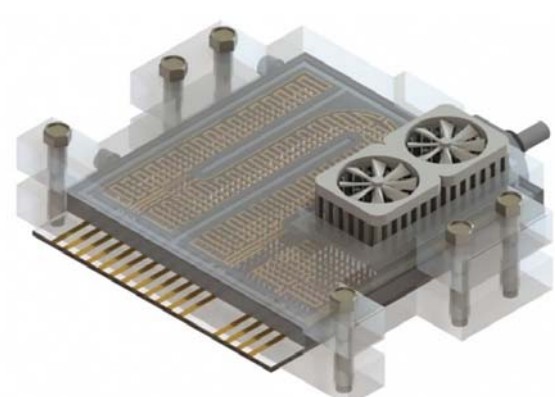

**(d) Attached cooling modules**

**Figure 3: Simplified fabrication process of the MEMS-based CPC.**

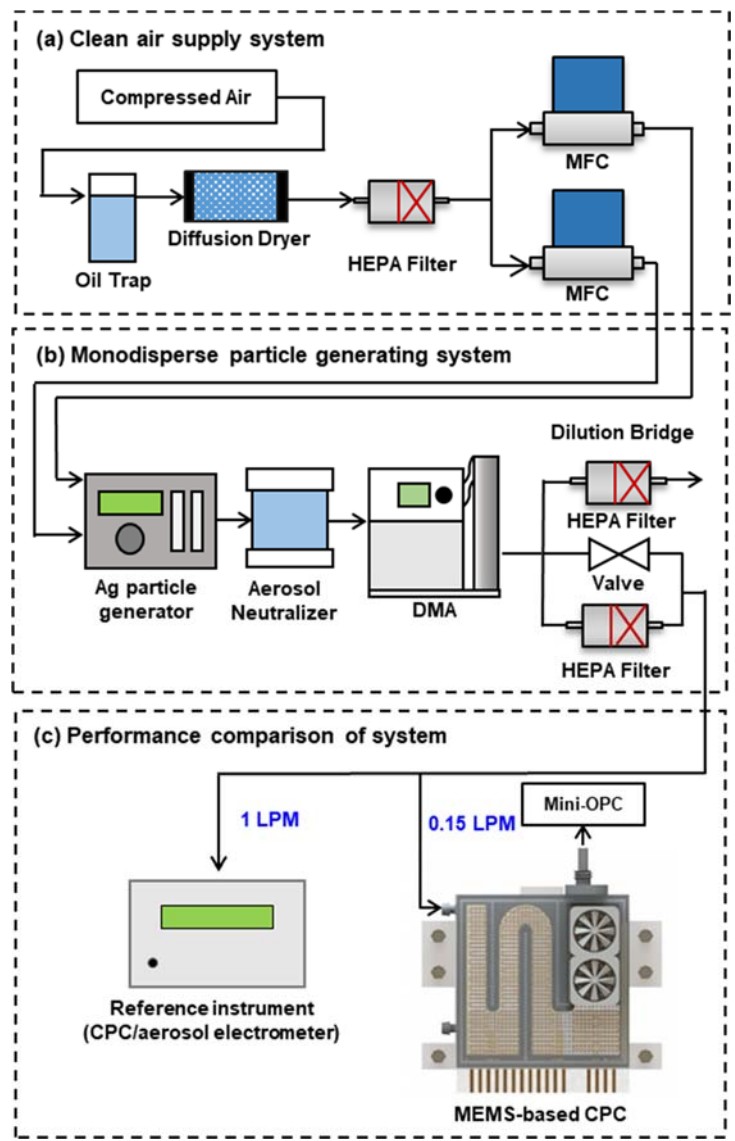

**Figure 4: Schematic of the experimental setup for evaluating the performance of the MEMS-based CPC.**

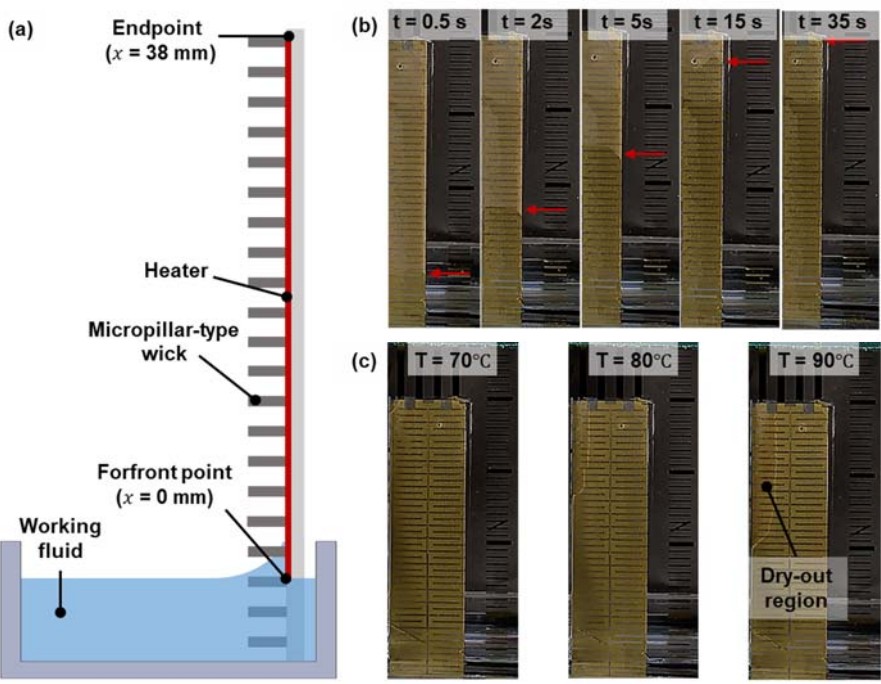

**Figure 5: (a) Schematic of the capillary rise experimental setup; (b) selected video frames from the rise of the working fluid using micropillar-type wick; (c) the dry-out region formation as the surface temperature increased.**

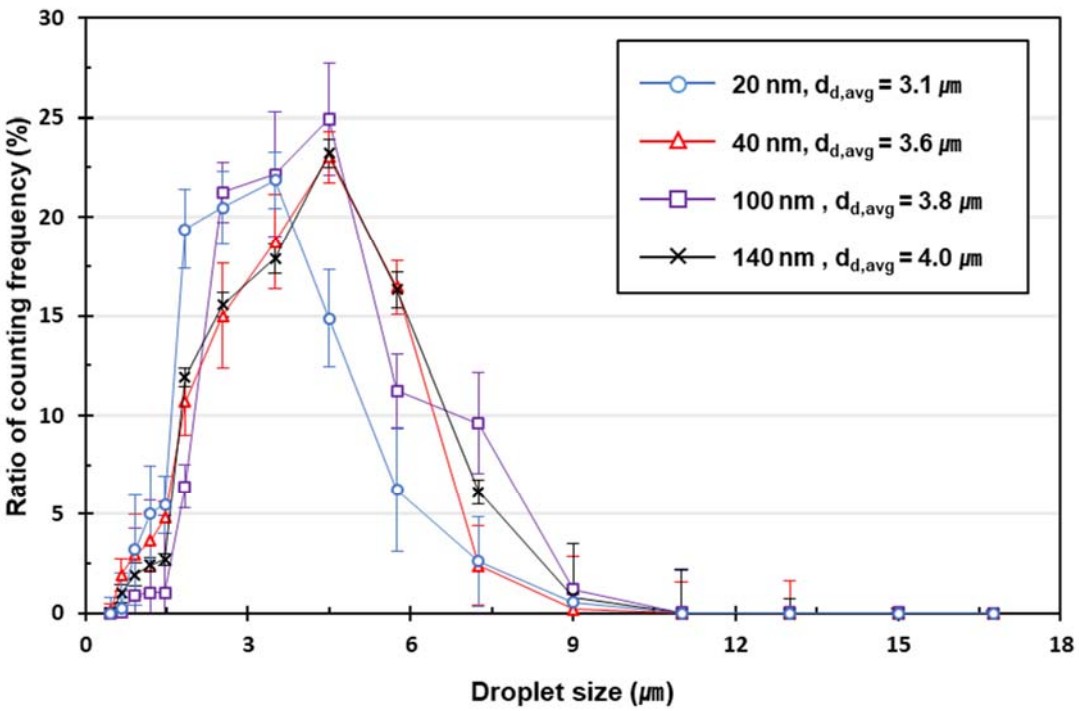

**Figure 6: Size distribution of the droplets grown from the MEMS-based particle growth system when Ag particles of specific sizes were introduced.**

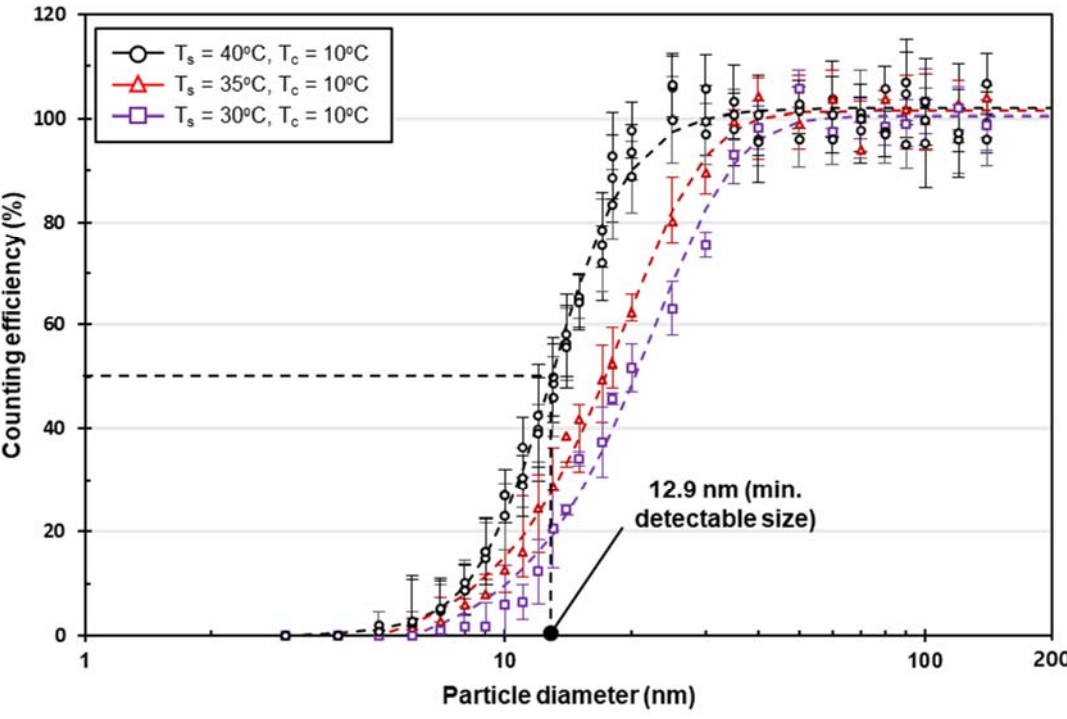

5 **Figure 7: Particle counting efficiency of the MEMS-based CPC as a function of the particle size and saturator temperature. The particle size at which the particle counting efficiency is 50 % was 12.9 nm (Ts = 40 °C), 17.3 (Ts = 35 °C), and 20.4 (Ts = 30 °C), respectively.**

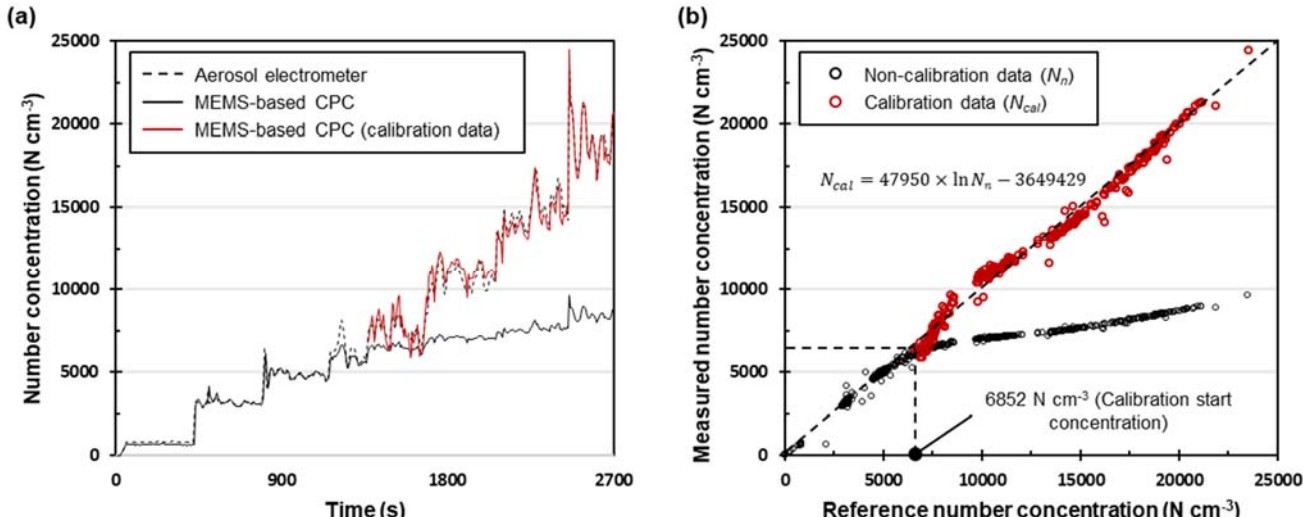

**Figure 8: (a) Time series of the number concentrations with the MEMS-based CPC and aerosol electrometer (b) one-to-one comparison of the measured number concentrations for both systems.**

| | Diameter (nm) | 28 | 26 | 24 | 22 | 20 | 18 | 16 |
|---|---|---|---|---|---|---|---|---|
| **Poisson statistics** | Time interval (s) | 96 | 66 | 84 | 72 | 108 | 108 | 66 |
| | Uncertainty (%) | 0.57 | 0.56 | 0.12 | 0.086 | 0.038 | 0.023 | 0.034 |
| **Number concentration (N cm$^{-3}$)** | Reference CPC | 8.36 | 18.99 | 223.84 | 585.87 | 1370.69 | 3733.96 | 4129.93 |
| | MEMS-based CPC | 7.99 | 17.65 | 224.58 | 619.10 | 1399.14 | 3809.45 | 4544.82 |
| | Difference (%) | 4.42 | 7.60 | -0.32 | -5.37 | -2.03 | -1.98 | -9.12 |

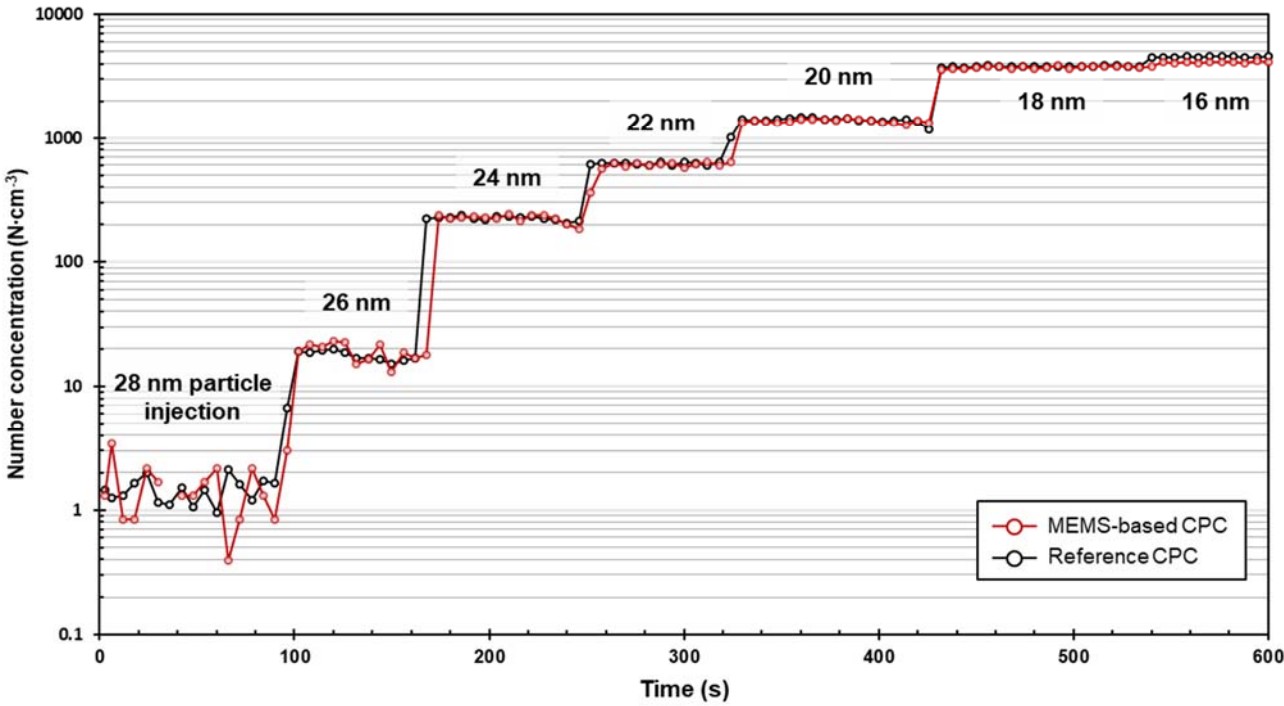

5 **Figure 9: Time series of the number concentrations measured with our system and reference CPC when the concentration and size were varied.**

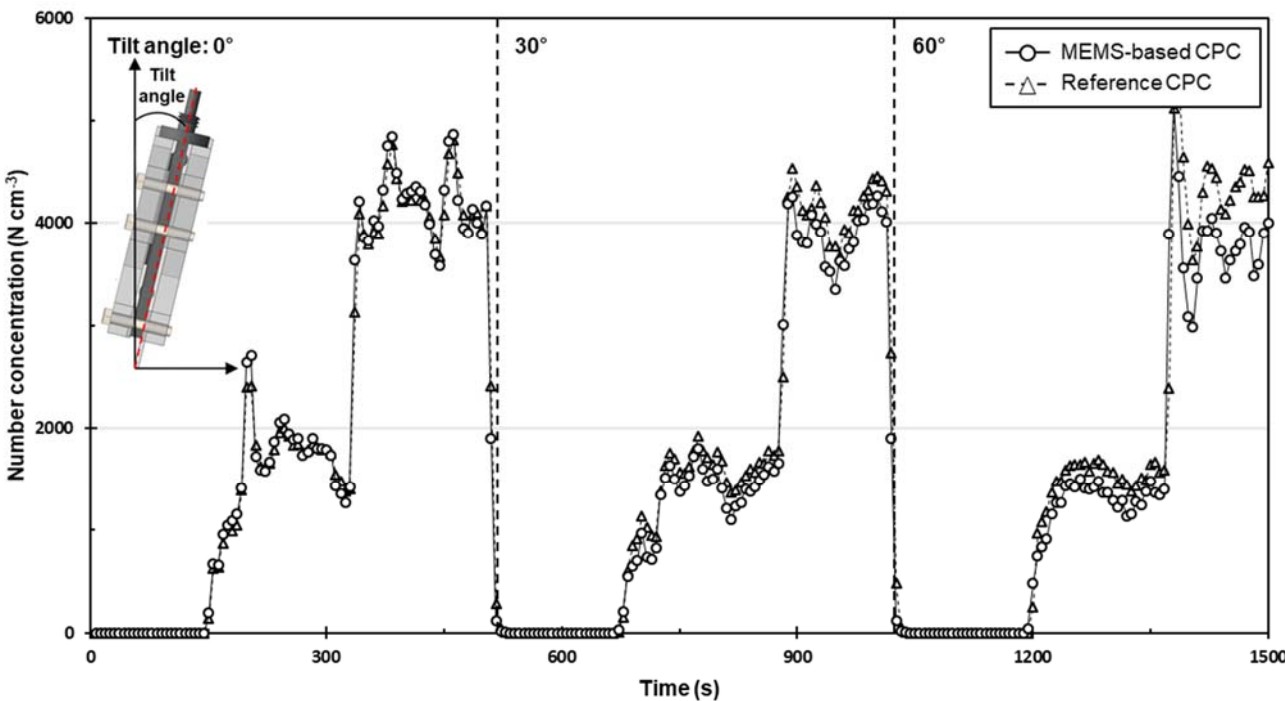

**Figure 10: Time series of the number concentrations measured by our system when it was tilted.**