# Peer review of "Microelectromechanical system-based condensation particle counter for real-time monitoring of airborne ultrafine particles"

_Atmospheric Measurement Techniques, 2019_

## Referee Comment (RC1) · Anonymous Referee #1 · 7 Mar 2019

General:

The manuscript of Yoo et al. presents design of a MEMS based CPC, its manufacturing process, and cut-off and concentration response characterization. The CPC design, to my knowledge, is unique and certainly deserves publication in AMT. Compared to commercial CPCs, the MEMS CPC reduces the size and weight significantly. The description of the manufacturing process seems adequate although outside of my expertize. Some claims are made on the cost-effectiveness of the CPC, which, however, should be considered carefully until the CPC is actually sold, as the final price of the CPC depends on many things, which are not discussed here, such as production

volumes, company structure etc. Possibly material costs can be compared if you get that information from other manufacturers, which I doubt. Indeed, I believe this manufacturing method can possibly be cheaper than the current designs, while the current manuscript does not support that with numbers.

The experimental characterization of the CPC is adequate, while the authors do not consider carefully enough why the cut-off is 13 nm. For example, according to its manual, TSI 3775 has temperatures of 39C in the saturator and 14C in the condenser, compared to 40C and 10C in the MEMS CPC. With smaller dT, the cut-off of the 3775 is 4 nm, while the MEMS CPC cut-off is 13 nm. TSI 3772 has cut-off of 10 nm at dT of 17C. How come? Even the Kelvin diameter inside the MEMS CPC is calculated to be 2.45 nm. Satisfactory explanation for this should include discussion and/or experiments and/or modeling of particle losses inside the CPC, the capability of the saturator to fully saturate the flow, and the condenser supersaturation profile. This should help to understand why the cut-off is 13 nm and not 2.45 nm what the Kelvin diameter predicts. Indeed, in careful experiments it has been found that the Kelvin diameter overestimates observed the cut-off, e.g. Iida et al. (2009) and Winkler et al. (2008).

Minor comments

P1 l18, i would not consider particle concentration of 7000 cm-3 as "high concentration" environment. The concentrations range from 10^0 in clean environments to 10^6-7 in very polluted or industrial applications, so 10^4 somewhere in the middle

P1 l20-25, There are various sources of UFPs, such as secondary particle formation in the atmosphere from the existing gases. Also their fraction of the total concentration is highly specific to the environment, or measurement time

P1 l29-30, "High-precision industries with cleanrooms also need UFP monitoring to increase the production yield", what does this mean? How is UFP and production yield connected?

[Figure]

P1 l34-36, why does it need to be portable or low-cost? Any "normal" CPC will give the same information

P2 l1-2, "because they are theoretically capable of counting every single UFP", this could be reformulated a little bit. Why e.g. your CPC is limited to concentration around 7000 cm-3 or smallest size of 13 nm? Same theoretical limitations apply for other CPC designs, just resulting in different limiting numbers.

P2 l3-4, "64 size channels per decade", from where this number is obtained? With a DMA you can in practice select almost infinite number of size channels

P2 L39, is the OPC also homemade or commercially available?

P4 l15-17, what is the difference between TSI Co. Ltd. and TSI Inc.?

Section 5.1, it is not evident whether there was a flow above the butanol surface when the dry-out region was measured. If not, does the flow have any effect on the dry-out region formation?

P5 l17-20, reformulate the sentences, partly badly written, partly ambiguous. "Initially" refers to A happening before B. The next sentence is ambiguous, what is the super-saturation that you calculate the Kelvin diameter? There is a supersaturation profile in the condenser, so single value for the Kelvin diameter is not good, unless it is a particle trajectory weighted average or something similar. Also 5 and 9 nm are quite much above the calculated Kelvin diameter, so why at 9 nm it increases sharply and not above 5 nm?

P5 l23, I would say non-negligible. If the Kelvin diameter is 2.45 nm, how come 90% of particles are detected only at 20 nm?

P5 l30, please indicate the background count rate also in units of count every x seconds. If these background counts are originating from homogeneous nucleation, it should already hint you why the cut-off of your CPC is so far away from the Kelvin prediction

P5 l30-32, reformulate. Temperature profile does not necessarily have anything to do with homogeneous background counts. It is true that homogeneous nucleation takes place at regions of high saturation ratio, but the temperature does not matter in case the saturation ratio is high enough.

P5 l32-P6 l6, does the concentration ratio CPC/AEM plateau immediately above concentrations of 7000 cm-3? With some corrections the CPC is possibly usable also at concentrations higher than 7000 cm-3.

P6 l10, why averaging of 6s? is the CPC performance comparable at 1s resolution?

P6 l22, what is the price of your CPC? Or manufacturing price (materials or materials+work) compared to manufacturing price of some other commercial CPC? I would be careful in making claims about cost-effectiveness without these numbers

P6 l23, 91.5% of what? Price, weight, volume?

Fig1, something missing from Air inlet (0.15

Fig4, why in the picture of aerosol electrometer reads condensation particle counter, and in the reference CPC reads electrometer?

References

Iida, K., Stolzenburg, M. R., McMurry, P. H. (2009). Effect of Working Fluid on Sub-2 nm Particle Detection with a Laminar Flow Ultrafine Condensation Particle Counter. Aerosol Sci Tech 43:81-96.

Winkler, P. M., Steiner, G., Vrtala, A., Vehkamaki, H., Noppel, M., Lehtinen, K. E. J., Reischl, G. P., Wagner, P. E., Kulmala, M. (2008). Heterogeneous nucleation experiments bridging the scale from molecular ion clusters to nanoparticles. Science 319:1374-1377.

---

## Referee Comment (RC2) · Anonymous Referee #2 · 8 Mar 2019

This manuscript reports on a small and seemingly simple condensation particle counter. The work appears sound, but the reporting fails the basic standard of scientific discourse - that the work be described in sufficient detail that a knowledgable researcher in the field could reproduce the work based upon the information provided in the manuscript. Key information is omitted from the paper that is essential to understand the work performed. It is not publishable in its present form, but could likely be made publishable with more thorough documentation. I also note that the statistics of the measurements need to be more thoroughly documented.

The CPC that is reported is a low flow (0.15 LPM) instrument that interfaces a clever,

3-D printed saturator and condenser with what appears to be a simple, off-the-shelf optical particle counter. No information is provided about the detector, which is a key component of the instrument, so I can only guess what it is. My suspicion is that this is one of the class of optical dust sensors that have been used as PM2.5 surrogate sensors. Such a sensor might yield quantitative data in this application, if the performance could be demonstrated to be stable over time. I note, however, that the flow paths through the instruments of this kind that I have examined are ill defined, and that the view volume and portion of the sample flow that is passed through the view volume are poorly constrained.

Therefore, a critical requirement for further consideration of this manuscript is explicit identification of the optical detector (both manufacturer and model number), or detailed description if the authors have developed it as part of their instrument, and clear and complete documentation of how the flow from the condensational activation core of the CPC is interfaced with the optical detector. If this can be done, the authors may be reporting on an instrument that solves a very real need for an affordable CPC. I note that the sketch in Fig. 1 shows a more elaborate OPC design than in the simple optical detectors about which I have read or with which I have worked. Moreover, it shows a forward scattering instrument, but does not show a beam dump that is required for this geometry, suggesting that the cartoon does not represent the actual instrument.

I further note that no mention is made of the pump employed, or the approach used for quantitative flow control. The instrument should be documented fully. A critical requirement for publication in any reputable scientific journal is full disclosure. I note that reports on the performance of commercialized instruments may fit within the scientific literature without such detail because they are are used by many groups and such documentation informs those users of issues in their use and the interpretation of the data they generate. Incomplete reports on a one-of-a-kind prototype instruments have little value and should not be published.

I further note that the authors have not identified the working fluid employed in their

[Figure]

CPC. I suspect that they have copied existing CPCs and used butanol, but have no way of knowing for sure from the information provided. This should be a very easy oversight to correct.

All of the data presented should be accompanied by explicit statement of the operating conditions in sufficient detail that the experiments could be reproduced. The temperature difference between the saturator and condenser is incomplete without specifying one of those two temperatures. Again, this is a small revision that should be easy for the authors to address.

The authors note that the CPC grows particles into 3.16 $\mu$m droplets, but provide no hint as to how they have determined this size. The instrumentation that they report is incapable of directly measuring particles of this size. I suspect that they have made the inference from the scattered light intensity detected by their OPC, but OPC measurements in this size range are highly uncertain owing to Mie resonances. A precision of 3 significant figures is highly unlikely.

The precision reported for accuracy of counts also seems excessive. A claim is made that the concentration accuracy is 4.1%, and later that it agrees with a reference CPC within 91.5%, suggesting double the uncertainty. Error bars are needed on the data plots, and uncertainties on the quoted efficiencies. The uncertainties in the reported efficiencies need to take into account the Poisson counting statistics. The counting time is another piece of information that needs to be documented. It is quite reasonable to operate with a longer counting time than the TSI instrument with which they compare their data - as a trade-off in producing a much smaller instrument.

The orientation of the instrument also needs to be specified, as well as its sensitivity to orientation. I suspect that it must be operated with the saturator oriented vertically, with the reservoir at the bottom. What is the sensitivity to motion or tilting, as this is likely with a small, highly portable instrument? Also, if the optics are protected from flooding with working fluid, that would be of interest in ascertaining the instrument's suitability

for different applications.

Specific points:

The introduction makes a range of broad statements about the importance of ultrafine particles in different situations, but these are based mostly upon two-decade-old reports. I do not dispute the importance of measuring such particles, but the specific arguments made have been superseded by more recent studies. The first paragraph of the introduction needs to be written with appropriate reference to the current understanding rather than that of two decades ago. The reference to clean room particle measurements is problematic because the flow rates through a miniature CPC will lead to poor counting statistics at the concentrations at which clean rooms operate. The introduction needs to be rewritten, but the amount of explanation of the need for simpler and cost-effective instrumentation can be reduced significantly, however.

On p. 2, l. 13 reference is made to electrical techniques. I can only guess to what technology the authors are referring. If they wish to refer to a specific technology, they must define it in sufficient detail that the reader can understand it without having to trace through the literature citations that they give.

l. 35. The authors claim a lower detection limit on concentration of 8 particles/cm$^3$. At the quoted flow rate, this concentration corresponds to about 20 counts/s. For a 1 s integrating time, this corresponds to a statistical (based on Poisson statistics) uncertainty of order 1/4. To claim uncertainty of 4%, a long count integration time is required. The authors need to be explicit about the parameters of the measurement, i.e., what is the integration/counting time?

On p. 3, l. 18, the authors note that a customized circuit was implemented for pulse-width-modulation. It would be highly desirable to document the nature of the control algorithm employed: is it P (proportional), PI (proportional-integral), or PID (proportional-integral-differential). Was the control algorithm implemented in software or in analog, hardware circuitry?

p. 4: The experimental setup section describes the system in reasonable detail. It is unclear, how the quantitation was performed. The data presented in Fig 6 shows an asymptotic approach to 100% detection efficiency. Were no large particles lost or otherwise not counted. Again, the nature of the optical detector and the way that the flow interfaces to it becomes an issue; How does the design ensure that all activated and grown particles pass through the view volume so that they can be counted. The analysis of the experimental results needs to clearly document all assumptions and approximations made in the data analysis. How were losses measured?

P. 5, l. 28: The authors report on the detectable concentration range as measured with the temperature difference between the condenser and saturator set at 30$^o$C. Specify the conditions of operation of the CPC fully. The temperature difference between the saturator and the condenser are only part of the information that is needed. What was was the saturator temperature? Of course, all temperature information is meaningless unless the authors specify their working fluid.

Fig. 5 is very difficult to interpret as there is little contrast difference between the region where liquid is present and where the wick is dry. A bit of guidance as to how to interpret the picture is appropriate.

Fig. 6. The data shown reveal three groupings of data points, one below 10 nm, one 10-15 nm, and one for larger particles, but only up to 40 nm. What was changed for these data sets that show slight offsets and differences in noise levels? Data are shown down to 3 nm which is not possible with the DMA that the authors report using. Did they use a different DMA for the sub-10 nm particles? The calibration should be extended to larger particles since the tortuous path through the saturator could lead to losses, and the small dimensions might make those losses orientation dependent.

[Figure]

---

## Referee Comment (RC3) · Anonymous Referee #3 · 16 Apr 2019

**Review of the AMT manuscript amt-2019-78**
"MEMS-based condensation particle counter for real-time monitoring
of airborne ultrafine particles at a point of interest"
by S.-J. Yoo et al., 2019

The above manuscript deals with description and characterization of a miniaturized Condensation Particle Counter (CPC). As CPCs are widely used in aerosol science and small and light CPCs were lacking in the past, the new approach is highly welcome and important. The manuscript is well structured and I have only comments which should be addressed by the authors to improve the manuscript.

**General remarks:**

The manuscript is well structured, however the last two paragraphs of the introduction read like a summary. They forestall important results from the following sections. The introduction is not the right place for this information. Please change and shorten these last two paragraphs. My suggestion, instead of presenting results, you should write something like:

"Traditional CPC geometries do not allow for a much smaller size and weight … they are not suited for batch production … we tried a new technique … based on …"

This would fit perfect to the end of the introduction.

But this is just an suggestion.

**Specific remarks:**

Title:

- Please write-out "MEMS", as this abbreviation is relatively unknown to the atmospheric community.

- Please remove "at a point of interest" because this statement I not very specific and a user of this new instrument will always only measure "at a point of interest". Moreover, I´m not a native speaker, but it seems for me that this wording might be understood as "sight-seeing point" as well.

Same for the abstract, remove "at a point of interest" as well; there it also causes a reference error, the "point of interest" is not "portable, …".

Abstract:

- p. 1, l. 15: Please specify if the given size information (nm) are for particle "diameter" or "radius". Please do so in the whole manuscript.

- p. 1, l. 16: Please specify which "deviation" is meant, standard deviation?

Introduction:

- p. 1, l. 22: "Monitoring of airborne ultrafine particles …yield(s !) enhancement in industry fields"
I can imagine what the authors meant with this sentence, but actually I do not understand it. Please rephrase.

- p. 1, l. 22: Some reference in the introduction seem for me to be very old, many 199Xs. There might be some very fundamental among those, but the last 20 years definitely brought some progress. Please check if there are newer references.

- p. 1, l. 23: "UFPs are mainly generated from burning fossil fuels …". This statement is not generally true, the main particle formation process is gas to particle formation and fossil fuel burning might be dominant in cities only.

- p. 2, l. 6: Please insert "e.g." before mentioning the TSI 3007". There are other models as well, e.g. from KANOMAX.

CPC Description

- p. 2, l. 35: I might have missed it, but you don´t specify in the whole manuscript which working fluid was used. Did you try different ones? This information is essential for this technical paper and should be provided to the reader.

- p. 2, l. 40: In the supplement is only one figure. It seems strange to me to have a supplement because of just one figure. I suggest to incorporate it into the main manuscript.

If I remember correctly, in the traditional CPCs the particles grow more or less to the same droplet size, but in this new type there is a strong increase in droplet size with initial particle diameter (Fig. S1). How far does this go? Are even 20 or 30 µm droplets generated? How does this affect the counting efficiency of your new CPC for larger particles (sedimentation)? Please comment in the manuscript on that, if this is an issue.

- p. 3, l. 19: I did not fully understand what the micropillars in the condenser do. They are needed to prevent droplet formation, which could clog the channel, fine, but how exactly do they do this? Please explain more in detail.

- p. 3, l. 22: Why is the pitch between the micropillars in the condenser not provided as number? All other dimensions are provided.

- p. 3, l. 29: If the Reynolds number in the system is so low (below 32) don´t you get problems with secondary flows, i.e. convection? Could you check this for instance using the Richardson number? How are the diffusional losses for such a flow? Please provide some numbers.

Experimental setup

- p. 4, l. 20: How long were the sampling lines? Which flow splitter was used? How is the flow geometry there? Did you have the same volume flow to all instruments (probably not, see Fig. 4)? A flow splitter can introduce strong deviations in the particle number concentration for different instruments connected to the flow splitter, in particular when using different

volume flows, because the particles are not necessarily distributed homogenously over the sampling line. Please add the information on how the flow split exactly looks like and how you guaranteed that all three instruments good the same particle number concentration. According to Fig 4c I would guess for very small particles this was not the case.

Results and discussion

- p. 4, l. 37: The "dry out-region" (maybe better "dry out region"?), how were they identified? The red areas in Fig. 5 "show" them, but I see no difference in the photo inside the read areas and outside.

- p. 5, l. 7: The activation efficiency is described as "the condensation chip at growing droplets" which I do not understand or feel to say something wrong. The activation efficiency is the fraction of particles being activated to droplets in the condensation chip.

- p. 5, l. 9: Please add "in particular for small particles below ca. 30 nm" after "… on particle size," because the mentioned dependencies are mainly valid for this range.

- p. 5, l. 16: Fig, 6, how often was the counting efficiency curve measured? The day to day slightly different set-up can influence the curve, hence it should be measured at least three times, ideally on different days. How were the temperature settings and how does the counting efficiency change with different temperature settings?

- p. 5, l. 20: "diffusivity of particles is inversely proportional to the size", is this true? How about the slip correction which brings a non-linear term into the particle diffusion problem?

- p. 5, l. 24: Again, how large do the droplet in your CPC get at maximum and is sedimentation really no problem?

Figures

- Fig. 1: I believe the miniature OPC scheme is incomplete. I cannot imagine that this OPC works in the forward scattering mode without using a beam blocker. Is this really true?

- Fig. 5: Please add the unit "s" to the times provided above the photos.

- Fig. 6.: Please provide uncertainty bars to the plot.

**Technical corrections:**

- p. 1 1. 12: Please remove "the" before "3D".

- p. 1, l. 17: Please replace "range of" with "rang is", otherwise a verb would be missing.

- p. 1, l. 17: The correct CF unit for the particle number concentration should be "$1/cm^3$", without "N". Please correct in the whole manuscript.

- p. 1, l. 22: A space is missing in before the "("…. This occurs several times in the manuscript, please correct.

- p. 2, l. 7: Please remove "for ownership", this addition is not needed here.

- p. 2, l. 18: Please remove the comma after "chip"

- p. 2, l. 33: Please insert "to" before "grow".

- p. 3, l. 2: Please exchange "proposed" with a different word, e.g. "new". The MEMS CPC is existing, you do not "propose" it.

- p. 3, l. 7: Please insert "micropillar" before "dimensions".

- p. 3, l. 18: Please insert "to" before "control".

- p. 3, l. 19: "… some" what? "may condense on the wall", please specify that you mean the working fluid vapor.

- p. 4, l. 14: Please exchange "They" with "The particles".

- p. 5, l. 4: Please delete "drawn"

- p. 5, l. 38: "the lower concentration limit of the aerosol electrometer was relatively high" Please rephrase, there is no "lower concentration limit" for an electrometer, however, because of the electronic noise you cannot trust the measured concentrations below a few hundred particles per cubic centimeter.

- p. 7: The format of some references are different compared to the others, e.g., Hajjam et al, 2010 or Kim et al., 2015. There are more, please check all.

---

## Author Comment (AC1) · 31 May 2019

Dear reviewer,

The reviewer's comments were highly insightful and enabled us to improve the quality of our manuscript. Our point by point responses to the each of the comments in the following pages. We hope that the revisions in the manuscript and our accompanying responses will be sufficient to make our manuscript suitable for publication in *Atmospheric Measurement Technique*.

**Changes to the revised manuscript are shown in red.**

We shall look forward to hearing from you at your earliest convenience.

Yours sincerely,

Professor Yong-Jun, Kim.

Address: School of Mechanical Engineering, Yonsei University, Seoul, Korea
Phone: +82-2-2123-7212
Fax: +82-2-312-2159
E-mail: yjk@yonsei.ac.kr

**OPEN DISCUSSION #2**

**Question 1**

No information is provided about the detector, which is a key component of the instrument, so I can only guess what it is. My suspicion is that this is one of the class of optical dust sensors that have been used as PM2.5 surrogate sensors. Such a sensor might yield quantitative data in this application, if the performance could be demonstrated to be stable over time.

**Answer 1**

We thank for your advice. Although the main argument of the manuscript is the chip-sized particle growth chip, there is little information about the detector.

The optical detector used in the proposed system is a detection part of the commercial optical particle counter (Innoair-615D, Innociple Co., KR) which is capable of counting individual particles larger than 0.3 μm (Fig. R1).

[Figure]

Figure R1: The specification of the commercial optical particle counter. Its detection part was used in this study.

Figure R2 shows the (a) exploded view, (b) section A-A' and (c) section B-B' of the optical detector. It consists of the sensing chamber and optics (laser, cylindrical lens, elliptic mirror, optical detector, light trap). Introduced droplets are firstly arranged in a row in the acceleration nozzle (i.e., the outlet of the particle growth chip) and enter the sensing chamber. The droplets then pass through the place where the condensed thin beam is irradiated. The mirror collects the scattered light from a droplet and redirect it to the optical detector.

When the laser beam passes through the cylindrical lens, the shape of the laser beam is not a point but a very thin surface. In addition, the acceleration nozzle at the chip outlet is only 0.8 mm in diameter and is located about 1.5 mm below the point where the beam passes. Therefore, as shown in Figure R2 (c), on the condition that the coincidence error does not occur (when two particles do not pass through the viewing volume at the same time), almost all the grown micro-droplets are counted in the optical detector.

In the perspectives of the structure and detection principle, the optical detector used in this study is similar to the high-precision OPC rather than dust sensors. Thus the proposed system demonstrated particle counting performance, which was comparable to those of the reference CPC (model 3772, TSI Inc., USA).

[Figure]

Figure R2: The (a) exploded view, (b) section A-A' and (c) section B-B' of the optical detector used in this study

**We added the description of the optical detector in '2 Description of the MEMS-based CPC' of the revised manuscript as following,**

~. The maximum Reynolds number in the channel at the given flow rate was only 32, which means that the sample stream in the channel was in the fully laminar regime.

The detection part of the commercial optical particle counter (OPC; Innoair-615D, Innociple Co., KR), which provided the time resolution of 6 s, was used as the optical detector in our system. It consists of the sensing chamber and optics (laser, cylindrical lens, elliptic mirror, optical detector, light trap). Introduced droplets are firstly arranged in a row in the acceleration nozzle (i.e., the outlet of the particle growth chip) and enter the sensing chamber. The droplets then pass through the place where the condensed thin beam is irradiated. The mirror directs the scattered light of a droplet to the sensing surface of the optical detector. When the laser beam passes through the cylindrical lens, the shape of the laser beam is not a point but a very thin surface. In addition, the acceleration nozzle at the chip outlet is only 0.8 mm in diameter and is located about 1.5 mm below the point where the beam passes. Therefore, on the condition that the coincidence error does not occur (the two particles do not pass through the viewing volume at the same time), almost all the grown micro-droplets are counted in the optical detector.

**Question 2**

I note that the sketch in Fig. 1 shows a more elaborate OPC design than in the simple optical detectors about which I have read or with which I have worked. Moreover, it shows a forward scattering instrument, but does not show a beam dump that is required for this geometry, suggesting that the cartoon does not represent the actual instrument.

**Answer 2**

We thank for letting us catch a mistake. We modified the schematic diagram of the miniature OPC based on the aforementioned mechanism in Answer 1.

**We modified the schematic diagram of the miniature OPC in 'Figure 1' of the revised manuscript as following,**

[Figure]

Figure 1: Schematic illustration of the MEMS-based CPC. The proposed system consists of four parts: the reservoir, saturator, condenser, and miniature OPC. The reservoir supplies the working fluid to the saturator via capillary action by the micropillar-type wick. The saturator heats the working fluid to generate saturated vapor. The saturated air becomes supersaturated when cooled by the condenser. UFPs grow into micro-sized droplets in the condenser and are counted by the miniature OPC.

**Question 3**

I further note that no mention is made of the pump employed, or the approach used for quantitative flow control. The instrument should be documented fully.

**Answer 3**

We thank for your advice and letting us improve the solidity of the manuscript. The MEMS-based CPC employed a micro pump (model 00H220H024, Nidec Co., JP) and a flow sensor (model FS1012-1020-NG, IDT Co., USA) for quantitative control. Furthermore, a lab-made circuit was used to regulate the flow rate based on the PID feed-back control. Moreover, we have described the customized circuit more clearly.

**We added the description of the flow control and customized circuit in the '2 Description of the MEMS-based CPC' of the revised manuscript as following,**

Figure 2a shows our system with the customized circuit. The circuit, whose dimension is 90 mm x 65 mm, simultaneously reads the data from the miniature OPC, temperature sensor and flow sensor (model FS1012-1020-NG, IDT Co., USA), and controls the power of the heaters, cooling modules and micro pump (model 00H220H024, Nidec

Co., JP) via a pulse-width-modulation (PWM) method. In order for our system to be a stand-alone device, the feedback loops based on the proportional-integral-differential (PID) algorithm is implemented in the micro control unit (MCU) of the circuit, and their gains can be easily controlled using serial communication.

5    **We added the optical photograph of the customized circuit as Figure 2a in the revised manuscript as followings,**

[Figure]

Figure 2: (a) Optical image of the proposed system; (b) magnified image of the heaters, resistive temperature sensors, and wick on the glass slide; and (c) scanning electron microscope (SEM) image of the micropillar-type wick.

**Question 4**

10    I further note that the authors have not identified the working fluid employed in their CPC.

**Answer 4**

We thank for letting us catch the mistakes in our manuscript. The kind of working fluid should be clearly stated in manuscript. We used Butanol as the working fluid of MEMS-based CPC.

**We modified 'Description of the MEMS-based CPC' part of the revised manuscript as following,**

Figure 1 shows the operating principle of the proposed MEMS-based CPC, which consists of a reservoir, saturator, condenser, and miniature OPC. To generate supersaturated vapor and hence grow UFPs to micro-sized droplets, the proposed system utilizes a conductive cooling method. Butanol was used as working fluid. The saturator generates

20    saturated vapor by heating the wetted wall with the working fluid.

**Question 5**

The authors note that the CPC grows particles into 3.16 µm droplets, but provide no hint as to how they have determined this size. The instrumentation that they report is incapable of directly measuring particles of this size. I suspect that they

25    have made the inference from the scattered light intensity detected by their OPC, but OPC measurements in this size range are highly uncertain owing to Mie resonances. A precision of 3 significant figures is highly unlikely.

**Answer 5**

A miniaturized OPC (PSM-615D, Innociple, KR) was used to characterize the counting performance of the proposed system, whereas another type OPC (OPC-N2, Alphasense, UK) was used to measure the mean diameters of the grown droplets. Both the OPCs are commercially available. The reason for using two kinds of OPCs is that, although the model PSM-615D can count a single droplet, it was not calibrated in terms of particle size. The model OPC-N2 is capable of not only measuring particles from 0.4 to 17 µm, but also having similar performance to the reference OPC (PAS-1.108, Grimm Technologies) (Sousan et al., 2016).

We modified the manuscript to clearly specify the measurement route for characterizing droplet size distribution, and refer to the measurement uncertainty induced by the Mie resonance. Also, the digit number of the droplet diameter was decreased to 2.

We further performed to characterize the mean diameter of the grown droplets when Ag particles in the size range from 20 to 140 nm were introduced, and move the results from the supplemental material to the result section in the revised manuscript.

**We have added '5.1 Droplet size distribution' in the result of the revised manuscript as following,**

Figure 6 shows the size distribution of the droplets generated from the MEMS-based particle growth system. Monodisperse Ag particles in the size range from 20 to 140 nm were used as test aerosol and their number concentrations were fixed at around 2000 N cm$^{-3}$ by adjusting the valves of the dilution bridge. The sampling time for measuring each droplet distribution was 2 min, and the corresponding measurement uncertainty based on the Poisson statistics was 0.13 %. All the error bars at each data point represent the standard deviations. The commercial OPC (OPC-N2, Alphasense, UK) was used for measuring the droplet size distribution. It was reported that OPC-N2 was capable of not only measuring particles from 0.4 to 17 µm, but also having moderate counting performance compared to the reference OPC (PAS-1.108, Grimm Technologies) (Sousan et al., 2016). The measurement errors induced from the Mie resonance was not considered in this data. The average droplet diameter ($d_{d,avg}$) was 3.1 µm when particles with the size of 20 nm, slightly larger than the minimum detectable size (12.9 nm), were introduced. Since the lower detectable size of the optical detector in the proposed system was 0.3 µm, introduced particles successfully grew into micrometer-sized droplets which were large enough to be counted by optical means. It was noted that the mean droplet size did not vary significantly above 40 nm. Also, most of the grown droplets were smaller than 10 µm, indicating that tens of micrometer-sized droplets, which could be attached to the inner walls of the particle growth system or optical detector via sedimentation, were barely generated.

[Figure]

Figure 6: The size distribution of the droplets grown from the MEMS-based particle growth system when Ag particles of specific sizes were introduced.

Sousan, S., Koehler, K., Hallett, L., and Peters, T. M.: Evaluation of the Alphasense Optical Particle Counter (OPC-N2) and the Grimm Portable Aerosol Spectrometer (PAS-1.108), Aerosol Sci Technol, 50, 1352-1365, 2016

**Question 6**

The precision reported for accuracy of counts also seems excessive. A claim is made that the concentration accuracy is 4.1%, and later that it agrees with a reference CPC within 91.5%, suggesting double the uncertainty.

**Answer 6**

We thank for your advice and letting us modifying the ambiguous part. The value of 91.5 % reported in the conclusion of the manuscript represents that the difference in physical volume of the proposed system and commercial portable CPC (model 3007, TSI Inc., USA).

**Avoiding the ambiguity of this value, we have modified the related sentence in '6 Conclusion' of the revised manuscript as following,**

Therefore, we modified the sentence. In terms of compactness and cost-efficiency, the proposed system is superior to conventional instruments. The physical volume of our system is only 8.5 % of the volume of the commercially-available portable CPC (e.g., model 3007, TSI Inc., USA).

**Question 7**

The orientation of the instrument also needs to be specified, as well as its sensitivity to orientation. I suspect that it must be operated with the saturator oriented vertically, with the reservoir at the bottom. What is the sensitivity to motion or tilting, as this is likely with a small, highly portable instrument? Also, if the optics are protected from flooding with working fluid, that would be of interest in ascertaining the instrument's suitability for different applications.

**Answer 7**

In order to minimize the loss of droplets in the condenser via sedimentation, it is recommended that the MEMS-based CPC be oriented perpendicular to the surface. However, as advised by the reviewer, since the proposed system has been developed as a portable device for on-site monitoring of UFPs, its measurement performance should be evaluated when the device is tilted.

The design for preventing working fluid in the reservoir from flooding into the optics has not been done yet, which will be addressed in the future studies.

**We have added the related experiment in '5.5 Performance comparison with the reference CPC' of the revised manuscript as followings,**

Figure 10 shows the measurement results of our system when it was tilted like an inset image. Monodisperse Ag particles with 25 nm were introduced and their concentrations were step-wisely increased from 0 to 4000 N cm$^{-3}$. Since

the measurement was carried out for about 500 s at each angle, and the measurement uncertainty of each section was below 0.01 %. When the proposed system was oriented perpendicular to the surface, the counting efficiency of the proposed system was 2.04 % which was similar to the result of the size-dependent counting efficiency. When a 30 ° angle was applied, the counting efficiency was 7.07%. At 60°, the measurement difference compared to the reference CPC exceeded 10 % (16.3 %). Thus, it was found that, at a tilt angle of 60° or less, MEMS-based CPC can monitor UFPs without the significant degradation of the accuracy.

The deviation of the counting efficiency induced from applying a tilt angle can be explained by the sedimentation of droplets in the condenser. At 0 °, since the gravity direction was identical to the direction of the sample flow, the probability that grown droplets impacted on the condenser wall via sedimentation was negligible. However, with the increment of the tilt angle, the velocity vector of a droplet perpendicular to the channel increased, which lead to the decrement of the counting efficiency.

[Figure]

Figure 10: The time series of the number concentrations measured by the proposed system when it was tilted.

**Question 8**

The first paragraph of the introduction needs to be written with appropriate reference to the current understanding rather than that of two decades ago.

**Answer 8**

Thank you for your advice. We added recent references to the first paragraph of the introduction.

**We modified 'Introduction' part of the revised manuscript as following,**

Monitoring of airborne ultrafine particles (UFPs), which are smaller than 100 nm, is needed in various fields for human health and yield enhancement in industrial fields(Donaldson et al., 1998; Donovan et al., 1985; Hristozov and Malsch, 2009; Li et al., 2016; Liu et al., 2015). UFPs are mainly generated from burning fossil fuels and are ubiquitous in urban air; they account for about 90% of the total particle number concentration(Kim et al., 2011; Kittelson, 1998; Shi et al., 1999). Because of dramatic developments in nanotechnology, engineered UFPs for commercial and research purposes have been produced at a large scale. These incidentally and intentionally generated UFPs are more harmful to human health than larger counterparts: UFPs have a higher chance to deposit in the lower respiratory system and are more toxic owing to their larger surface-to-volume ratios, which causes oxidative stress, pulmonary inflammation, and tumor

development(Hesterberg et al., 2012; Hext, 1994; Li et al., 2003; Renwick et al., 2004). Thus, onsite monitoring is needed to assess and minimize UFP exposure. High-precision industries with cleanrooms also need UFP monitoring to increase the production yield. For instance, in the semiconductor industry, the minimum linewidth of the chips is approaching 7 nm(Neisser and Wurm, 2015). Particles that are a few nanometers in size are critical because "killer particles" (i.e., the diameter is greater than half of the minimum linewidth) can render the whole chip unusable(Libman et al., 2015). Unfortunately, since UFPs in cleanrooms are generated during fabrication processes (e.g., chemical vapor deposition (CVD), metallization, wet etching), contamination can occur in any manufacturing stages(Choi et al., 2015; Manodori and Benedetti, 2009). In these circumstances, a portable and low-cost sensor is needed for onsite UFP monitoring to accurately evaluate adverse health effects and control the contamination level in cleanrooms to enhance the production yield.

**We have added the references in the 1st paragraph of the introduction in the revised manuscript.**

(1) Li, N., Georas, S., Alexis, N., Fritz, P., Xia, T., Williams, M. A., Horner, E., and Nel, A.: A work group report on ultrafine particles (American Academy of Allergy, Asthma & Immunology): Why ambient ultrafine and engineered nanoparticles should receive special attention for possible adverse health outcomes in human subjects, J Allergy Clin Immunol, 138, 386-396, 2016.

(2) Liu, L., Urch, B., Poon, R., Szyszkowicz, M., Speck, M., Gold, D. R., Wheeler, A. J., Scott, J. A., Brook, J. R., Thorne, P. S., and Silverman, F. S.: Effects of ambient coarse, fine, and ultrafine particles and their biological constituents on systemic biomarkers: a controlled human exposure study, Environ Health Perspect, 123, 534-540, 2015.

(3) Kim, K. H., Sekiguchi, K., Kudo, S., and Sakamoto, K.: Characteristics of Atmospheric Elemental Carbon (Char and Soot) in Ultrafine and Fine Particles in a Roadside Environment, Japan, Aerosol and Air Quality Research, 11, 1-12, 2011.

(4) Hesterberg, T. W., Long, C. M., Bunn, W. B., Lapin, C. A., McClellan, R. O., and Valberg, P. A.: Health effects research and regulation of diesel exhaust: an historical overview focused on lung cancer risk, Inhal Toxicol, 24 Suppl 1, 1-45, 2012.

**Question 9**

The reference to clean room particle measurements is problematic because the flow rates through a miniature CPC will lead to poor counting statistics at the concentrations at which clean rooms operate.

**Answer 9**

You have raised an important point. Flow rate is an important factor that determines the quality of counting statistics in low concentration environment such as clean room.

Cleanrooms are classified according to the number and size of particles permitted per volume of air. In order to reduce the costs of the maintenance, the semiconductor industries separate one cleanroom facility into multiple classes; a cleanroom ranges from class 1 to class 100,000 depending on the processes taking place within the facility.

Clean rooms with class 1~10 require CPCs (e.g., Aerotrak[R] 9001, TSI Inc., USA) with a high flow rate (2.83 LPM) to monitor the excessively low background concentrations. However, in cleanrooms with class 100 or 1000, the number concentration of background particles is in the order of 1 N cm$^{-3}$. In this case, although the proposed system operates at low volumetric flow rate (0.15 LPM = 2.5 cm$^3$ s$^{-1}$), the measurement uncertainty based on the Poisson distribution reach 10 % in 40 s. Thus, it is expected that the proposed system can monitor the background concentration in cleanrooms with a satisfactory resolution. For example, Liao et al. also successfully measured the number concentration of backgrounds

$(0.2 \sim 22.41$ N cm$^{-3}$) in a cleanroom using a portable CPC (model 3007, TSI Inc., USA) with a flow rate of 0.1 LPM (Liao et al., 2018).

**We added a relevant discussion in the '5.4 Detectable concentration range' of the revised manuscript as following,**

Even if filtered air is introduced into a system, droplets may form in the condenser via homogeneous or ion-induced nucleation. Droplets without UFP nuclei cause false counting, which makes the system read a higher concentration than reality. This phenomenon is critical, especially in low-concentration environments. To evaluate the false counting of the proposed system, it was operated for 1 h with a HEPA filter connected to its inlet. When the temperature difference between the saturator and condenser was set to 30 °C, the average number concentration during the measurement period (background concentration) was only 0.05 N cm$^{-3}$. This result indicated that homogeneous nucleation hardly occurred. Thus, the temperature profile was uniformly established inside the condensation chip because homogeneous nucleation typically occurs at low temperatures in regions where the local saturation ratio is high. Owing to the low false count performance, our system can be applied to monitoring UFPs in cleanrooms with class 100 or 1000 whose background concentrations are in order of 1 N cm$^{-3}$ (Liao et al., 2018). In these environment, the measurement uncertainty based on the Poisson statistics is expected to be 10 % in a sampling time of 40 s for a given flow rate (0.15 LPM = 2.5 cm$^3$ s$^{-1}$).

**We have added the references in the '5.4 Detectable concentration range' of the revised manuscript.**

Liao, B.-X., Tseng, N.-C., Li, Z., Liu, Y., Chen, J.-K., and Tsai, C.-J.: Exposure assessment of process by-product nanoparticles released during the preventive maintenance of semiconductor fabrication facilities, Journal of Nanoparticle Research, 20, 2018.

**Question 10**

On p. 2, l. 13 reference is made to electrical techniques. I can only guess to what technology the authors are referring. If they wish to refer to a specific technology, they must define it in sufficient detail that the reader can understand it without having to trace through the literature citations that they give.

**Answer 10**

We thank for your advice and letting us improve the solidity of the manuscript. The electrical method in the manuscript is the particle detection method where the number concentration of UFPs is measured by electrically charging them and sensing their current. Following your suggestions,

**We modified the '2nd paragraph of Introduction' in the revised manuscript as following,**

Condensation particle counters (CPCs) are one of the most widely used UFP detection instruments and are based on the heterogeneous particle condensation technique(Stolzenburg and McMurry, 1991). They grow UFPs to micro-sized droplets through condensation and capable of counting every single UFP by optical means. Compared to an electrical method (measuring the number concentration of UFPs by electrically charging them and sensing their current), CPCs provide extremely sensitive and precise counting because they are capable of counting because they are capable of counting individual particles. (Kangasluoma et al., 2017; Kangasluoma et al., 2014; McMurry, 2000).

**Question 11**

Error bars are needed on the data plots, and uncertainties on the quoted efficiencies. The uncertainties in the reported efficiencies need to take into account the Poisson counting statistics. The counting time is another piece of information that needs to be documented. It is quite reasonable to operate with a longer counting time than the TSI instrument with which they compare their data - as a trade-off in producing a much smaller instrument.

l. 35. The authors claim a lower detection limit on concentration of 8 particles/cm3. At the quoted flow rate, this concentration corresponds to about 20 counts/s. For a 1 s integrating time, this corresponds to a statistical (based on Poisson statistics) uncertainty of order 1/4. To claim uncertainty of 4%, a long count integration time is required. The authors need to be explicit about the parameters of the measurement, i.e., what is the integration/counting time?

**Answer 11**

The measurement uncertainties were specified in the results (droplet size distribution, size-dependent particle counting efficiency, detectable concentration range and performance comparison with the reference CPC) of the proposed system. Since the values of the uncertainty and standard errors were very small (in order of 0.1 %), the standard deviations were used as error bars for each result.

**We have added the uncertainty in the ʻ5.2 Droplet size distribution' of the revised manuscript.**

Figure 6 shows the size distribution of the droplets generated from the MEMS-based particle growth system. Monodisperse Ag particles in the size range from 20 to 140 nm were used as test aerosol and their number concentrations were fixed at around 2000 N cm$^{-3}$ by adjusting the valves of the dilution bridge. The sampling time for measuring each droplet distribution was 2 min, and the corresponding measurement uncertainty based on the Poisson statistics was 0.13 %. All the error bars at each data point represent the standard deviations.

[Figure]

Figure 6: The size distribution of grown droplets grown from the MEMS-based condensation chip when Ag particles of specific sizes were introduced.

**We have added the uncertainty and error bars in the ʻ5.3 Size-dependent particle counting efficiency' of the revised manuscript.**

Figure 7 shows the size-dependent counting efficiency of the MEMS-based CPC. The size range of Ag particles was controlled concentration range to 1000-2000 N cm$^{-3}$. The sampling times for each data point were 300 s, and the measurement uncertainty based on the Poisson statistics was 0.02%.

[Figure]

Figure 7: Particle counting efficiency of the MEMS-based CPC as a function of the particle size and saturator temperature. The particle size at which the particle counting efficiency was fitted to 50% was 12.9 nm (TS = 40oC), 17.3 (TS = 35) and 20.4 (TS = 30), respectively.

**We have added the measurement uncertainty in the '5.5 Performance comparison with reference' of the revised manuscript.**

Figure 10 shows the measurement results of our system when it was tilted like an inset image. Monodisperse Ag particles with 25 nm were introduced and their concentrations were step-wisely increased from 0 to 4000 N cm$^{-3}$. Since the measurement was carried out for about 500 s at each angle, and the measurement uncertainty of each section was below 0.01 %.

**We have added the measurement uncertainty in the 'Figure 9' of the revised manuscript.**

| Diameter (nm) | | 28 | 26 | 24 | 22 | 20 | 18 | 16 |
|---|---|---|---|---|---|---|---|---|
| **Poisson statistics** | Time interval (s) | 96 | 66 | 84 | 72 | 108 | 108 | 66 |
| | Uncertainty (%) | 0.57 | 0.56 | 0.12 | 0.086 | 0.038 | 0.023 | 0.034 |
| **Number concentration (N cm$^{-3}$)** | Reference CPC | 8.36 | 18.99 | 223.84 | 585.87 | 1370.69 | 3733.96 | 4129.93 |
| | MEMS-based CPC | 7.99 | 17.65 | 224.58 | 619.10 | 1399.14 | 3809.45 | 4544.82 |
| | Difference (%) | 4.42 | 7.60 | -0.32 | -5.37 | -2.03 | -1.98 | -9.12 |

[Figure]

Figure 9: Time series of the number concentrations measured with the proposed system and reference CPC when the concentration and size were varied.

**Question 12**

On p. 3, l. 18, The authors note that a customized circuit was implemented for pulse-width-modulation. It would be highly desirable to document the nature of the control algorithm employed: is it P (proportional), PI (proportional-integral), or PID (proportional-integral-differential). Was the control algorithm implemented in software or in analog, hardware circuitry?

**Answer 13**

Please refer to Answer 3.

**Question 13**

p. 4: The experimental setup section describes the system in reasonable detail. It is unclear, how the quantitation was performed. The data presented in Fig 6 shows an asymptotic approach to 100% detection efficiency.

Were no large particles lost or otherwise not counted? Again, the nature of the optical detector and the way that the flow interfaces to it becomes an issue; How does the design ensure that all activated and grown particles pass through the view volume so that they can be counted. The analysis of the experimental results needs to clearly document all assumptions and approximations made in the data analysis. How were losses measured?

Fig. 6. The data shown reveal three groupings of data points, one below 10 nm, one 10-15 nm, and one for larger particles, but only up to 40 nm. What was changed for these data sets that show slight offsets and differences in noise levels? The calibration should be extended to larger particles since the tortuous path through the saturator could lead to losses, and the small dimensions might make those losses orientation dependent.

**Answer 13**

Because of the large difference between the flow rates of the commercial reference instrument and MEMS-based CPC, the following procedures were carried out to verify that particles with the same concentration were introduced into the two systems. First, to minimize the particle loss induced from the turbulence at the bifurcation, a flow splitter with a very small angle of cleavage (model 3708, TSI Inc., USA) was used. The tubes which leads to the both systems were electrostatic dissipative to minimize the electrostatic particle loss, and their lengths were carefully adjusted to match the transportation times.

To verify that the particles which introduced into both systems have the same concentrations, it was confirmed that the counting efficiency was close to 100 % when particles with size of 100 nm were introduced (it was assumed that they were activated and grew into droplets with 100 % efficiency). Then, while reducing the size of the introduced particles to 40 nm by adjusting the voltages of a DMA, it was confirmed whether the counting efficiency remained constant. Through these procedures, it was verified that the concentrations of the particles delivered to the two systems were the same.

The loss of the proposed system was characterized using the counting efficiency, since it is defined as the efficiency of the system at detecting the introduced particles, and thereby describes the overall transportation/activation efficiencies.

The size-dependent counting efficiency of the proposed system was further characterized using Ag particles in the size range of 3 to 140 nm (Figure 7). Particles with size of 140 nm were almost the maximum size that the Ag particle generator (EP-NGS20, EcoPictures, Co. KR) could generate. The number concentration range of the Ag particles was controlled to be 1000–2000 N cm$^{-3}$. Although there are small fluctuations, the counting efficiency of the particles larger than 25 nm was nearly close to 100 % ($T_S$ = 40ºC), indicating that the loss of large particles or count missing of the OPC were negligible. These results proved that the OPC virtually counted all activated and grown particles which passed through its sensing zone. The reason why the OPC can count almost all the droplets is discussed in Answer 1.

**We modified 'Size-dependent particle counting efficiency' part of the revised manuscript as following,**

Figure 7 shows the size-dependent counting efficiency of the MEMS-based CPC. The size range of Ag particles was controlled concentration range to 1000-2000 N cm$^{-3}$. The sampling times for each data point were 300 s, and the measurement uncertainty based on the Poisson statistics was 0.02%. To evaluate the effect of the temperature difference, the counting efficiency was characterized when the condenser temperature ($T_c$) was 10 º C and the saturator temperatures ($T_s$) were 30, 35 and 40 º C. At 40 ºC (the design value of the saturator temperature), the same experiments were repeated three times to confirm the measurement reliability. When the saturator temperature was 40 º C, it was found that our system detected 1% of UFPs with the size of 5 nm, and the detection efficiency increased sharply above 9 nm. This was primarily because the activation efficiency ($\eta_{act}$) increased when the particle size exceeded the Kelvin diameter (2.34 nm). The transport efficiency ($\eta_{trans}$) also increased, because the diffusivity of a particle decreases with the increment of the particle size. The counting efficiency data were curve-fitted using

$$\eta_d = \alpha + \frac{(\beta - \alpha)}{1 + (d_p/\gamma)^\delta} , \qquad\qquad (2)$$

where α, β, γ and δ are fitting constants of 101.96, 2.00, 12.99 and 4.70, respectively. The corresponding minimum detectable size is defined as the size at which particles are detected with 50% efficiency and was found to be 12.9 nm. The detection efficiency was 90% at 20.1 nm and reached 95% at 22.9 nm. It was close to 100 % and constant in the size range from 25 to 140 nm, indicating that the internal particle loss in this size range was negligible.

[Figure]

Figure 7: Particle counting efficiency of the MEMS-based CPC as a function of the particle size and saturator temperature. The particle size at which the particle counting efficiency was fitted to 50% was 12.9 nm (TS = 40oC), 17.3 (TS = 35) and 20.4 (TS = 30), respectively.

**We added the procedures and assumptions for characterizing the counting efficiency in 'Experimental setup' part of the revised manuscript as following,**

Because of the large difference between the flow rates of the reference instrument and our system, the following procedures were carried out to verify that particles with the same concentration were introduced into the two systems. First, to minimize system the particle loss induced from the turbulence at the bifurcation, a flow splitter with a very small angle of cleavage (model 3708, TSI Inc., USA) was used. The tubes which leads to the both systems were electrostatic dissipative to minimize the electrostatic particle loss, and their lengths were carefully adjusted to match the transportation times. To verify that the particles which introduced into both systems have the same concentrations, it was confirmed that the counting efficiency was close to 100 % when particles with size of 100 nm were introduced (it was assumed that they were activated and grew into droplets with 100 % efficiency). Then, while reducing the size of the introduced particles to 40 nm by adjusting the voltages of a DMA, it was confirmed whether the counting efficiency remained constant. Through these procedures, it was verified that the concentrations of the particles delivered to the two systems were the same. The loss of our system was characterized using the counting efficiency, since it is defined as the efficiency of the system at detecting the introduced particles, and thereby describes the overall transportation/activation efficiencies.

**Question 14**

P. 5, l. 28: The authors report on the detectable concentration range as measured with the temperature difference between the condenser and saturator set at 30 ºC. Specify the conditions of operation of the CPC fully. The temperature difference between the saturator and the condenser are only part of the information that is needed. What was the saturator temperature? Of course, all temperature information is meaningless unless the authors specify their working fluid.

**Answer 14**

The temperatures of the saturator and condenser were 40 °C and 10 °C, respectively that we have written on page 2 of the original manuscript: "The saturated sample then enters a condenser, whose temperature (10 °C) is lower than that of the saturator (40 °C)."

**Question 15**

Fig 5 is very difficult to interpret as there is little contrast difference between the region where liquid is present and where the wick is dry. A bit of guidance as to how to interpret the picture is appropriate.

**Answer 15**

**We modified the video frames in 'Figure 5' of the revised manuscript as following,**

[Figure]

Figure 5: (a) Schematic of the capillary rise experimental setup; (b) selected video frames from the rise of the working fluid using micropillar-type wick; (c) the dry-out region formation as the surface temperature increased.

**Question 16**

Data are shown down to 3 nm which is not possible with the DMA that the authors report using. Did they use a different DMA for the sub-10 nm particles?

**Answer 16**

Thank for your correction. We used Ag particles in the size range from 3 to 140 nm. We classified the particles using two DMAs depending on their sizes: (1) nano DMA (model 3085, TSI Co. Ltd., USA) from 3 - 10 nm, (2) long DMA (model 3081A, TSI Co. Ltd., USA) from 11 - 140 nm.

[Figure]

Figure R3: The (a) nano DMA and (b) long DMA used in this study.

**We modified 'Experimental setup' part of the revised manuscript as following,**

[revised manuscript text omitted]

---

## Author Comment (AC2) · 3 Jun 2019

Dear reviewer,

The reviewer's comments were highly insightful and enabled us to improve the quality of our manuscript. Our point by point responses to the each of the comments in the following pages. We hope that the revisions in the manuscript and our accompanying responses will be sufficient to make our manuscript suitable for publication in *Atmospheric Measurement Technique*.

Changes to the revised manuscript are shown in blue.

We shall look forward to hearing from you at your earliest convenience.

Yours sincerely,

Professor Yong-Jun, Kim.

Address: School of Mechanical Engineering, Yonsei University, Seoul, Korea
Phone: +82-2-2123-7212
Fax: +82-2-312-2159
E-mail: yjk@yonsei.ac.kr

**OPEN DISCUSSION #1**

**Question 1**

P1-18, I would not consider particle concentration of 7000 cm$^{-3}$ as "high concentration" environment. The concentrations range from $10^0$ cm$^{-3}$ in clean environments to $10^{6-7}$ cm$^{-3}$ in very polluted or industrial applications, so $10^4$ cm$^{-3}$ somewhere in the middle.

**Answer 1**

We thank for your advice and agree with you. The number concentration range where the MEMS-based CPC can singly count particles was characterized as 7.99 ~ 6850 cm$^{-3}$. The high concentration in the original manuscript meant the upper concentration limit of our system, which was subjective term. Thus, we have deleted this term.

**We modified the 'Abstract' of the revised manuscript as following,**

Our system measured the UFP number concentration with high accuracy (mean difference within 4.1 %), and the number concentration range where the proposed system can singly count particles was characterized as 7.99–6850 cm$^{-3}$. Thus, the proposed system has a potential of being used for UFP monitoring in various environments (e.g., air filtration system, high-precision industries utilizing cleanrooms, indoor/outdoor atmospheres).

**Question 2**

P1 l20-25, There are various sources of UFPs, such as secondary particle formation in the atmosphere from the existing gases. Also their fraction of the total concentration is highly specific to the environment, or measurement time.

**Answer 2**

We thank for your advice and letting us improve the solidity of the manuscript. We have modified the text on the sources of UFPs in the revised manuscript, referring to the newly-added references.

**We modified the 1st paragraph of 'Introduction' in the revised manuscript as following,**

Monitoring of airborne ultrafine particles (UFPs), which are smaller than 100 nm, is needed in various fields for human health and yield enhancement in industrial fields (Donaldson et al., 1998; Donovan et al., 1985; Hristozov and Malsch, 2009; Li et al., 2016; Liu et al., 2015). While they have a variety of anthropogenic and natural sources, in urban area, UFPs are largely generated from the vehicle exhaust (e.g., soot agglomerates, secondary particles from hazardous gaseous precursors) (Kim et al., 2011; Kittelson, 1998; Shi et al., 1999). Moreover, because of dramatic developments in nanotechnology, engineered UFPs for commercial and research purposes have been produced at a large scale. These incidentally and intentionally generated UFPs are more harmful to human health than larger counterparts: UFPs have a higher chance to deposit in the lower respiratory system and are more toxic owing to their larger surface-to-volume ratios, which causes oxidative stress, pulmonary inflammation, and tumor development (Hesterberg et al., 2012; Hext, 1994; Li et al., 2003; Renwick et al., 2004). Thus, onsite monitoring is needed to assess and minimize UFP exposure.

**Question 3**

P1 l29-30, "High-precision industries with cleanrooms also need UFP monitoring to increase the production yield", what does this mean? How is UFP and production yield connected?

**Answer 3**

Currently, semiconductor devices are being processed at the linewidth of a few nanometers. Thus, the related manufacturing process requires an extremely clean environment to prevent contamination. Although the air-purification system with ultra-low particulate (ULPA) filter eliminate the contaminants of the air entering the clean room, it cannot control the internally-generated UFPs the manufacturing instruments under operation (e.g., chemical vapor deposition (CVD), metallization, wet etching). As shown in Figure R1, these UFPs can be attached to wafers during the manufacturing process. If they are deposited on electrodes of a chip, they cause the interruption of the current flow, making the whole chip unusable and thereby reducing the yield of the semiconductor device. In this regard, ISO 14644-12, "Cleanroom and associated controlled environment: Specifications for monitoring air cleanliness by nanoscale particle concentration", has been developed to guide how to monitor UFPs in cleanrooms.

[Figure]

Figure R1: (a) the schematic view and (b) SEM images of UFP deposition on a wafer.

**We modified the 1st paragraph of 'Introduction' in the revised manuscript as following,**

High-precision industries with cleanrooms also need UFP monitoring to increase the production yield. For instance, in case of the semiconductor industry, as the minimum feature size of the semiconductor devices approaches to 7 nm, particles with the diameter of a few nanometers are critical (Neisser and Wurm, 2015). Although the air-purification system equipped with ultra-low particulate (ULPA) filter eliminate the contaminants in the air entering the clean room, it cannot control the internally-generated UFPs during the manufacturing processes (e.g., chemical vapor deposition (CVD), metallization, wet etching) (Choi et al., 2015; Manodori and Benedetti, 2009). If they are deposited on electrodes of a chip, they cause the interruption of the current flow, making the whole chip unusable and thereby reducing the yield of the semiconductor device (Libman et al., 2015). In this regard, ISO 14644-12 has been recently developed to guide how to monitor UFPs in cleanrooms.

**Question 4**

P1 l34-36, Why does it need to be portable or low-cost? Any "normal" CPC will give the same information.

**Answer 4**

You have raised an important point. Compared to commercial CPCs, the major advantages of the proposed system are compactness and cost efficiency, making us actively utilize it for on-site monitoring applications.

The typical CPCs (model 3025, TSI Inc., USA), which provide the accurate number concentration of UFPs as small as 3 nm, are mainly used for academic research due to their high cost and large size. Although, to my knowledge, the smallest portable CPC with the lower detectable size of 10 nm is currently in the market (model 3007, TSI Inc., USA), it is still considered bulky (292 mm × 140 mm × 140 mm) and expensive for the ownership (~ $ 10,000). In the contrast, the proposed system is far smaller and cheaper than even the portable CPC without the significant performance degradation (minimum detectable size: 12.9 nm).

The compactness and cost-efficiency of CPCs are preferred when monitoring in real-world environments where spatial and temporal variations of UFP concentration are enormous. For example, in urban areas, the sources of UFPs are highly localized and their migration patterns driven by air-flow are very complex, because skyscrapers, people and traffic are highly concentrated. Also, in cleanrooms, since a majority of UFPs are mainly generated from the manufacturing instruments under operations, their sources are also localized. Thus, it is required to establish simultaneous monitoring at multiple points or dense monitoring networks for measuring these environments. The proposed system has a potential of being an appropriate solution for these applications, because each sensing node must be not only accurate, but also compact and inexpensive.

**We modified the 1ˢᵗ paragraph of 'Introduction' in the revised manuscript as following,**

In order to monitor the concentration field of UFPs in these environments where the spatial and temporal variations of UFP concentrations are enormous, a portable and low-cost sensors are required to establish simultaneous monitoring at multiple points or establish dense monitoring networks.

**Question 5**

P2 ll-2, "because they are theoretically capable of counting every single UFP", this could be reformulated a little bit. Why e.g. your CPC is limited to concentration around 7000 cm$^{-3}$ or smallest size of 13 nm? Same theoretical limitations apply for other CPC designs, just resulting in different limiting numbers.

**Answer 5**

We thank for your advice and agree with you. Any confusing sentence in the manuscript must be corrected. The sentence, "they are theoretically capable of counting every single UFP", will be reformulated as they are capable of counting individual particles."

**We modified the 2ⁿᵈ paragraph of 'Introduction' in the revised manuscript as following,**

Condensation particle counters (CPCs) are one of the most widely used UFP detection instruments and are based on the heterogeneous particle condensation technique. They grow UFPs to micro-sized droplets through condensation and count them by optical means. Compared to an electrical method (measuring the number concentration of UFPs by electrically charging them and sensing their current), CPCs provide extremely sensitive and precise counting because they are capable of counting individual particles.

**Question 6**

P2 l3-4, "64 size channels per decade", from where this number is obtained? With a DMA you can in practice select almost infinite number of size channels

**Answer 6**

5      We thank for your advice. The DMA can select almost infinite number of size channels by adjusting its classification voltages. 64 size channel per decades means that the commercially-available scanning mobility particle sizer (SMPS) typically perform monitoring of UFP size distribution with a resolution of 64 channels per decade. However, since this phrase is confusing and the number of size channels of SMPS is not established, the '64 size channel per decades' terms will be deleted in the revised manuscript.

**We modified 'Description of the MEMS-based CPC' part of the revised manuscript as following,**

Moreover, if a differential mobility analyzer (DMA) is used as a particle size selector, CPCs can offer higher particle size resolution than any other particle-sizing instruments (Sioutas, 1999; Stolzenburg et al., 2017).

**Question 7**

15      P2 L39, is the OPC also homemade or commercially available?

**Answer 7**

We thank for your advice. Although the main argument of the manuscript is the chip-sized particle growth chip, there is little information about the detector.

The optical detector used in the proposed system is a detection part of the commercial optical particle counter (Innoair-
20      615D, Innociple Co., KR) which is capable of counting individual particles larger than 0.3 μm (Figure R2).

[Figure]

Figure R2: The specification of the commercial optical particle counter. Its detection part was used in this study.

25      Figure R3 shows the (a) exploded view, (b) section A-A' and (c) section B-B' of the optical detector. It consists of the sensing chamber and optics (laser, cylindrical lens, elliptic mirror, photodiode, light trap). Introduced droplets are firstly arranged in a row in the acceleration nozzle (i.e., the outlet of the particle growth chip) and enter the sensing chamber.

The droplets then pass through the place where the condensed thin beam is irradiated. The mirror collects the scattered light from a droplet and redirect it to the photodiode.

When the laser beam passes through the cylindrical lens, the shape of the laser beam at the focal point is not a point but a very thin surface. In addition, the acceleration nozzle at the chip outlet is only 0.8 mm in diameter and is located about 1.5 mm below the point where the beam passes. Therefore, as shown in Figure R3 (c), on the condition that the coincidence error does not occur (when two particles do not pass through the viewing volume at the same time), almost all the grown micro-droplets are counted in the optical detector.

In the perspectives of the structure and detection principle, the optical detector used in this study is similar to the high-precision OPC rather than dust sensors. Thus the proposed system demonstrated particle counting performance, which was comparable to those of the reference CPC (model 3772, TSI Inc., USA).

[Figure]

Figure R3: The (a) exploded view, (b) section A-A' and (c) section B-B' of the optical detector used in this study.

**We added the description of the optical detector in '2 Description of the MEMS-based CPC' of the revised manuscript as following,**

~. The maximum Reynolds number in the channel at the given flow rate was only 32, which means that the sample stream in the channel was in the fully laminar regime.

The detection part of the commercial optical particle counter (OPC; Innoair-615D, Innociple Co., KR), which provided the time resolution of 6 s, was used as the optical detector in the proposed system. It consists of the sensing chamber and optics (laser, cylindrical lens, elliptic mirror, photodiode, light trap). Introduced droplets are firstly arranged in a row in the acceleration nozzle (i.e., the outlet of the particle growth chip) and enter the sensing chamber. The droplets then pass through the place where the condensed thin beam is irradiated. The mirror directs the scattered light of a droplet to the sensing surface of the photodiode. When the laser beam passes through the cylindrical lens, the shape of the laser beam at the focal point is not a point but a very thin surface. In addition, the acceleration nozzle at the chip outlet is only 0.8 mm in diameter and is located about 1.5 mm below the point where the beam passes. Therefore, on the condition that the coincidence error does not occur (the two particles do not pass through the viewing volume at the same time), almost all the grown micro-droplets are counted in the optical detector.

**Question 8**

P4 l15-17, what is the difference between TSI Co. Ltd. and TSI Inc.?

**Answer 8**

We thank for letting us catch the mistakes in our manuscript. TSI Co. Ltd. and TSI Inc. are the same company. In the revised manuscript, we will unify with TSI Inc.

**We modified '4 Experimental setup' of the revised manuscript as following,**

They were electrically charged by a soft X-ray charger (XRC-05, HCT Co., KR) and then classified to a specific diameter with two types of DMA: (1) nano DMA (model 3085, TSI Co. Ltd., USA) for particles in the size range from 3 to 10 nm, (2) long DMA (model 3081A, TSI Co. Ltd., USA) for particles in the size range from 5 to 140 nm. Next, the number concentration of the monodisperse Ag particles were controlled (0–24000 N cm$^{-3}$) in the dilution bridge system by adjustment of the needle valve. Finally, the concentration-controlled and monodisperse Ag particles were introduced into the proposed system and reference instrument, which was either a CPC (model 3772, TSI Inc., USA) or aerosol electrometer (model 3068B, TSI Inc., USA).

**Question 9**

Section 5.1, it is not evident whether there was a flow above the butanol surface when the dry-out region was measured. If not, does the flow have any effect on the dry-out region formation?

**Answer 9**

**We described the flow condition and modified in '5.1 Working fluid transmission and evaporation' of the revised manuscript as following,**

In order to obtain optical images of the dry-out region formation, a single glass slide with the patterned electrodes and micropillar array was used. For this reason, there was no flow rate during the experiment. Although, due to the advection, the flow rate has effect on the area of the dry-out region to some extent, the vapor pressure at which the dry out region started to form (164.1 mmHg at 80 °C) was 8.7 times larger than the designed value of the saturator (18.9 mmHg at 40 °C). Thus, although there was no flow above the butanol surface when measuring the dry-out region, the dry-out region didn't occur in the saturator of the MEMS-based CPC under operating condition.

**Question 10**

P5 l17-20, reformulate the sentences, partly badly written, partly ambiguous. "Initially" refers to A happening before B.

P5 l23, I would say non-negligible. If the Kelvin diameter is 2.45 nm, how come 90 % of particles are detected only at 20 nm? The experimental characterization of the CPC is adequate, while the authors do not consider carefully enough why the cut-off is 13 nm. For example, according to its manual, TSI 3775 has temperatures of 39C in the saturator and 14C in the condenser, compared to 40C and 10C in the MEMS CPC. With smaller dT, the cut-off of the 3775 is 4 nm, while the MEMS CPC cut-off is 13 nm. TSI 3772 has cut-off of 10 nm at dT of 17C. How come? Even the Kelvin diameter inside the MEMS CPC is calculated to be 2.45 nm. Satisfactory explanation for this should include discussion and/or experiments and/or modeling of particle losses inside the CPC, the capability of the saturator to fully saturate the flow, and the condenser supersaturation profile. This should help to understand why the cut-off is 13 nm and not 2.45 nm

what the Kelvin diameter predicts. Indeed, in careful experiments it has been found that the Kelvin diameter overestimates observed the cut-off, e.g. Iida et al. (2009) and Winkler et al. (2008).

The next sentence is ambiguous, what is the supersaturation that you calculate the Kelvin diameter? There is a supersaturation profile in the condenser, so single value for the Kelvin diameter is not good, unless it is a particle trajectory weighted average or something similar. P5 l17-20, 5 and 9 nm are quite much above the calculated Kelvin diameter, so why at 9 nm it increases sharply and not above 5 nm?

**Answer 10-1**

We thank for letting us know the ambiguity of the original manuscript.

**We have modified the sentence in '5.3 Size-dependent particle counting efficiency' of the revised manuscript on as followings,**

Figure 7 shows the size-dependent counting efficiency of the MEMS-based CPC. The size range of Ag particles was controlled concentration range to 1000-2000 N cm$^{-3}$. The sampling times for each data point were 300 s, and the measurement uncertainty based on the Poisson statistics was 0.02 %. It was found that the proposed system detected 1 % of UFPs with the size of 5 nm, and the detection efficiency increased sharply above 9 nm.

**Answer 10-2**

[Figure]

Figure 7: Particle counting efficiency of the MEMS-based CPC as a function of the particle size and saturator temperature. The particle size at which the particle counting efficiency was fitted to 50 % was 12.9 nm (Ts = 40 ºC), 17.3 (Ts = 35 ºC) and 20.4 (Ts = 30 ºC), respectively.

The particle loss of our system was characterized using the counting efficiency, since it is defined as the efficiency of the system at detecting the introduced particles, and thereby describes the overall transportation/activation efficiencies. To accurately measuring the counting efficiency, the following procedures were carried out to verify that particles with the same concentration were introduced into the two systems. First, to minimize the particle loss induced from the turbulence at the bifurcation, a flow splitter with a very small angle of cleavage (model 3708, TSI Inc., USA) was used. The tubes which leads from the flow splitter to the both systems were electrostatic dissipative to minimize the electrostatic particle loss, and their lengths were carefully adjusted to match the transportation times. To verify that the particles which introduced into both systems have the same concentrations, it was confirmed that the counting efficiency was close to 100 % when particles with size of 100 nm were introduced (it was assumed that they were activated and grew into droplets with 100 % efficiency). Then, while reducing the size of the introduced particles to 40 nm by adjusting the voltages of a

DMA, it was confirmed whether the counting efficiency remained constant. Through these procedures, it was verified that the concentrations of the particles delivered to the two systems were the same. The size-dependent counting efficiency of the proposed system was further characterized using Ag particles in the size range of 3 to 140 nm (Figure 7). Particles with size of 140 nm were almost the maximum size that the Ag particle generator (EP-NGS20, EcoPictures, Co. KR) could generate. The number concentration range of the Ag particles was controlled to be 1000–2000 N cm$^{-3}$. Although there are small fluctuations, the counting efficiency of the particles larger than 25 nm was nearly close to 100 % ($T_S$ = 40$^o$C). It was close to 100 % and constant in the size range from 25 to 140 nm, indicating that the internal particle loss in this size range was negligible.

As the reviewer's advice, since the saturation ratio is the highest at the centreline in the condenser and close to 1 on the wall, the Kelvin diameter should be expressed as the profile in the condenser channel section rather than a single value. However, unfortunately, we did not have CFD software available to characterize the Kelvin distribution at present. Saturation ratio was calculated using butanol saturation vapor pressures at the wall temperatures of the saturator and condenser. The Kelvin diameter was obtained based on this value, which were seemed to be small value, because the temperature at the centreline of the condenser channel was higher than the wall temperature. Nonetheless, the reason why the minimum detectable size was relatively high for the given temperature difference can be explained. Since the saturator and condenser were close to each other and thereby have thermal interference, the condenser temperature might be higher than its originally-designed value due to the heat transfer from the saturator to the condenser. It is expected that this problem can be solved by increasing the thickness of the thermal barrier between the saturator and condenser.

**We modified '5.3 Size-dependent particle counting efficiency' of the revised manuscript as following,**

Figure 7 shows the size-dependent counting efficiency of the MEMS-based CPC. The size range of Ag particles was controlled concentration range to 1000-2000 N cm$^{-3}$. The sampling times for each data point were 300 s, and the measurement uncertainty based on the Poisson statistics was 0.02 %. To evaluate the effect of the temperature difference, the counting efficiency was characterized when the condenser temperature ($T_c$) was 10 $^o$C and the saturator temperatures ($T_s$) were 30, 35 and 40 $^o$ C. At 40 $^o$ C (the design value of the saturator temperature), the same experiments were repeated three times to confirm the measurement reliability. When the saturator temperature was 40 $^o$ C, it was found that our system detected 1 % of UFPs with the size of 5 nm, and the detection efficiency increased sharply above 9 nm. This was primarily because the activation efficiency ($\eta_{act}$) increased when the particle size exceeded the Kelvin diameter (2.34 nm). The transport efficiency ($\eta_{trans}$) also increased, because the diffusivity of a particle decreases with the increment of the particle size. The counting efficiency data were curve-fitted using

$$\eta_d = \alpha + \frac{(\beta - \alpha)}{1 + \left(d_p/\gamma\right)^\delta} \ , \tag{3}$$

where $\alpha$, $\beta$, $\gamma$ and $\delta$ are fitting constants of 101.96, 2.00, 12.99 and 4.70, respectively. The corresponding minimum detectable size is defined as the size at which particles are detected with 50 % efficiency and was found to be 12.9 nm. The detection efficiency was 90 % at 20.1 nm and reached 95 % at 22.9 nm. It was close to 100 % and constant in the size range from 25 to 140 nm, indicating that the internal particle loss in this size range was negligible.

The minimum detectable size was relatively higher than that of the commercial CPC operating at the same temperature difference. Since the saturator and condenser were close to each other and thereby have thermal interference, the condenser temperature might be higher than its originally-designed value due to the heat transfer from the saturator to the condenser. It is expected that this problem can be solved by increasing the thickness of the thermal barrier between the saturator and condenser.

**Question 11**

P5 l30, please indicate the background count rate also in units of count every x seconds. If these background counts are originating from homogeneous nucleation, it should already hint you why the cut-off of your CPC is so far away from the Kelvin diameter.

**Answer 11**

**We thank for your pointing. We modified '5.4 Detectable concentration range' of the revised manuscript as following,**

When the temperature difference between the saturator and condenser was set to 30 °C, the average number concentration and counting rate during the measurement period (background concentration) was 0.05 N cm$^{-3}$ and 0.125 N s$^{-1}$, respectively, indicating that homogeneous nucleation hardly occurred.

**Question 12**

P5 l32-P6 l6, does the concentration ratio CPC/AEM plateau immediately above concentrations of 7000 cm$^{-3}$? With some corrections the CPC is possibly usable also at concentrations higher than 7000 cm$^{-3}$.

**Answer 12**

You have raised an important point. Even if the number concentration is higher than the upper limit of the proposed system, it can be measured through calibration using curve fitting or nephelometric technique.

The proposed system was compared with an aerosol electrometer at a UFPs number concentration of 0 to 24000 N cm$^{-3}$. Figure 8 shows (a) the time series of number concentrations measured with the proposed system and an aerosol electrometer and (b) a one-to-one comparison of the measured number concentrations by both systems. When the UFP number concentration exceeded 7000 N cm$^{-3}$, the difference in the number concentration of the proposed system and the aerosol electrometer gradually increased. When the concentration exceeded 6852 N cm$^{-3}$, the logarithmic function was well fitted to the response curve of the proposed system, as expressed in Figure 8b. In case that the calibration was applied to the proposed system, the average difference between both systems was only 2.8 %.

**We modified 'Figure 8' of the revised manuscript as followings,**

[Figure]

**Figure 8: (a) Time series of the number concentrations with the MEMS-based CPC and aerosol electrometer; (b) one-to-one comparison of the measured number concentrations for both systems.**

We discussed the measurement accuracy in high concentration in '5.4 Detectable concentration range' of the **revised manuscript** as followings,

Thus, in case of the concentration exceeded 6852 N cm$^{-3}$, the logarithmic function was fitted to the response curve of the proposed system, as expressed in Figure 8b. When the calibration based on the fitted curve was applied to the proposed system, the average difference between both systems was within 2.8 %.

**Question 13**

P6 l10, why averaging of 6s? is the CPC performance comparable at 1s resolution?

**Answer 13**

The reason why the data was averaged over 6 s was that the purchased OPC, which was used for calibration, provided the options of 6s, 30s and 1 min resolution. If it operates at 1 s resolution, it is expected that the measurement uncertainty induced from Poisson statistics will be increased.

**We added the time resolution of the miniature OPC in '2 Description of the MEMS-based CPC' of the revised manuscript on page x as followings,**

The detection part of the commercial optical particle counter (OPC; Innoair-615D, Innociple Co., KR), which provided the time resolution of 6 s, was used as the optical detector in the proposed system.

**Question 14**

Some claims are made on the cost-effectiveness of the CPC, which, however, should be considered carefully until the CPC is actually sold, as the final price of the CPC depends on many things, which are not discussed here, such as production volumes, company structure etc. Possibly material costs can be compared if you get that information from other manufacturers, which I doubt. Indeed, I believe this manufacturing method can possibly be cheaper than the current designs, while the current manuscript does not support that with numbers.

I believe this manufacturing method can possibly be cheaper than the current designs, while the current manuscript does not support that with numbers. P6 l22, what is the price of your CPC? Or manufacturing price (materials or materials + work) compared to manufacturing price of some other commercial CPC? I would be careful in making claims about cost-effectiveness without these numbers.

**Answer 14**

Although it is necessary to determine the actual price of the proposed system, please understand that, since we are a graduate laboratory, it is difficult to calculate its exact price with numbers. The reason is that all the materials used in the proposed system were purchased at a retail price with minimum quantity and therefore relatively expensive. In addition, the MEMS process is based on a batch process, it has a great advantage in mass production. Therefore, in the present situation where its demo version has been just developed, it is difficult to calculate the accurate price including the depreciation cost of manufacturing equipment and labor cost, or considering the effect of company structure. Also, it is

difficult to compare the price of the proposed system with that of commercial CPC because of the limited information about specific material + work put into the production of commercial CPCs.

The main argument of this paper is the production of low-cost particle growth unit using MEMS technology. Therefore, in this paper, the materials used in the MEMS-based particle growth system will be listed in the supplemental information, and model number and manufacturer of other components (e.g., pumps, flow meters) will be specified in the revised manuscript.

**We added the component and price information in the supplemental information as followings,**

The materials used for the MEMS-based particle growth system are summarized in Table S1.

Table S1: Material used for manufacturing a single MEMS-based particle growth system.

| Components | | Used material | Manufacturer | Retail price | Used quantity | Price ($) |
|---|---|---|---|---|---|---|
| 3D printed channel | | SLA (Stereolithography Apparatus) | 3D MON, KR | 10 $ / EA | 1 | 10 |
| Integrated glass slide | Glass | Sodalime glass | SEMISTORE, KR | 3 $ / EA | 2 EA | 6 |
| | electrodes | AZ GXR-601 Photoresist | Microchem, USA | 420 $ / 3.8 L | 5 ml | 0.55 |
| | | MIF-300 developer | Microchem, USA | 170 $ / 20 L | 200 ml | 1.7 |
| | Micropillar wick | SU-8 2100 photoresist | Microchem, USA | 840 $ / 500 ml | 5 ml | 8.4 |
| | | SU-8 developer | Microchem, USA | 336 $ / 3.8 L | 100 ml | 8.8 |

**Question 15**

P6 l23, 91.5% of what? Price, weight, volume?

**Answer 15**

**We modified 'Conclusion' part of the revised manuscript on page 6 as following,**

In terms of compactness and cost-efficiency, the proposed system is superior to conventional instruments. The physical volume of the proposed system is only 8.5 % of the volume of the commercially-available portable CPC (e.g., model 3007, TSI Inc., USA).

**Question 16**

Fig1, something missing from Air inlet

Fig4, why in the picture of aerosol electrometer reads condensation particle counter, and in the reference CPC reads electrometer?

**Answer 16**

**Thank you for letting us know. We modified 'Figure 1' in the revised manuscript as following,**

[Figure]

Figure 1: Schematic illustration of the MEMS-based CPC. The proposed system consists of four parts: the reservoir, saturator, condenser, and miniature OPC. The reservoir supplies the working fluid to the saturator via capillary action by the micropillar-type wick. The saturator heats the working fluid to generate saturated vapor. The saturated air becomes supersaturated when cooled by the condenser. UFPs grow into micro-sized droplets in the condenser and are counted by the miniature OPC.

**We modified 'Figure 4' in the revised manuscript as following,**

[Figure]

Figure 4: Schematic of the experimental setup for evaluating the performance of the MEMS-based CPC.

---

## Author Comment (AC3) · 3 Jun 2019

Dear reviewer,

The reviewer's comments were highly insightful and enabled us to improve the quality of our manuscript. Our point by point responses to the each of the comments in the following pages. We hope that the revisions in the manuscript and our accompanying responses will be sufficient to make our manuscript suitable for publication in *Atmospheric Measurement Technique*.

**Changes to the revised manuscript are shown in green.**

We shall look forward to hearing from you at your earliest convenience.

Yours sincerely,

Professor Yong-Jun, Kim.

Address: School of Mechanical Engineering, Yonsei University, Seoul, Korea
Phone: +82-2-2123-7212
Fax: +82-2-312-2159
E-mail: yjk@yonsei.ac.kr

**OPEN DISCUSSION #3**

**Question 1**

The manuscript is well structured, however the last two paragraphs of the introduction read like a summary. They forestall important results from the following sections. The introduction is not the right place for this information. Please

5    change and shorten these last two paragraphs. My suggestion, instead of presenting results, you should write something like: "Traditional CPC geometries do not allow for a much smaller size and weight … they are not suited for batch production … we tried a new technique … based on …" This would fit perfect to the end of the introduction. But this is just a suggestion.

**Answer 1**

10    We thank for your kind advice.

**We modified the '1ˢᵗ paragraph of Introduction' in the revised manuscript as following,**

In this study, we developed a high-performance MEMS-based CPC that is portable, inexpensive, and power-efficient. Our system comprises a MEMS-based condensation chip and miniature optical particle counter (OPC). UFPs are grown

15    to micrometer-sized droplets on the chip and the grown droplets are detected by the miniature OPC. New fabrication techniques including MEMS and 3D printing technologies are applied to CPCs; particle enlargement (i.e., the fundamental process of the CPC) can be realized on a chip-scale system, because the essential elements for growing droplets (i.e., channels, micropillar-type wick, heaters, temperature sensors) are integrated on a single glass slide. Accordingly, our system shows far more compact and cost-efficient than traditional CPCs even including their portable versions.

20    In addition to its compactness, our system also provides high degrees of accuracy and precision. A quantitative characterization using Ag particles proves that our system is capable of growing UFPs to micrometer-sized droplets, counting them one by one, and thereby measuring UFP number concentration with a high accuracy, which is comparable to commercial OPC. These results show that our system can potentially be used as a portable, low-cost, and high-precision UFP sensor for various fields (e.g., assessing UFP exposure, monitoring workplaces, tracing particle sources in high-

25    precision industries with cleanrooms). Moreover, when combined with the recently developed miniature DMA, our system should also be able to perform onsite monitoring of the UFP size distribution with high resolution(Liu and Chen, 2016; Qi and Kulkarni, 2016).

**Question 2**

30    Title:

- Please write-out "MEMS", as this abbreviation is relatively unknown to the atmospheric community.

- Please remove "at a point of interest" because this statement I not very specific and a user of this new instrument will always only measure "at a point of interest". Moreover, I´m not a native speaker, but it seems for me that this wording might be understood as "sight-seeing point" as well.

35    Same for the abstract, remove "at a point of interest" as well; there it also causes a reference error, the "point of interest" is not "portable, …".

**Answer 2**

We thank for your advice. We wrote-out "MEMS" in the title, and removed the phrase, "at a point of interest", throughout the revised manuscript.

**Question 3**

Abstract:

- p. 1, l. 15: Please specify if the given size information (nm) are for particle "diameter" or "radius". Please do so in the whole manuscript.

- p. 1, l. 16: Please specify which "deviation" is meant, standard deviation?

**Answer 3**

Following your advices,

**We modified the terms 'Abstract' in the revised manuscript as following,**

**Abstract.** We present a micro-electro-mechanical systems (MEMS)-based condensation particle counter (CPC) for sensitive and precise monitoring of airborne ultrafine particles (UFPs) that is portable, inexpensive, and accurate. Our system consists of two main parts: a MEMS-based condensation chip that grows UFPs to micro-sized droplets and a miniature optical particle counter (OPC) that singly counts grown droplets with the light scattering method. A conventional conductive cooling–type CPC is miniaturized through MEMS technology and 3D printing technique, and the essential elements for growing droplets are integrated on a single glass slide. Our system is much more compact (75 mm × 130 mm × 50 mm), lightweight (205 g), and power-efficient (2.7 W) than commercial CPCs. In quantitative experiments, the results indicated that Our system can detect UFPs with a diameter of 13.4 nm by growing them to micro-sized (3.16 µm) droplets. Our system measured the UFP number concentration with high accuracy (mean difference within 4.1 %), and the number concentration range where our system can singly count particles was characterized as 7.99–6850 $cm^{-3}$. Thus, our system has a potential of being used for UFP monitoring in various environments (e.g., air filtration system, high-precision industries utilizing cleanrooms, indoor/outdoor atmospheres).

**Question 4**

p. 1, l. 22: "Monitoring of airborne ultrafine particles …yield(s !) enhancement in industry fields"

I can imagine what the authors meant with this sentence, but actually I do not understand it. Please rephrase.

**Answer 4**

We thank for your advice. We have rephrased the sentence and added a detailed explanation in the manuscript to help the reader understand.

**We modified the '1st paragraph of Introduction' in the revised manuscript as following,**

High-precision industries with cleanrooms also require UFP monitoring to trace their sources, minimize the contamination, and thereby increase the production yield. For instance, in case of the semiconductor industry, as the

minimum feature size of the semiconductor devices approaches to 7 nm, particles with the diameter of a few nanometers are critical (Neisser and Wurm, 2015). Although the air-purification system equipped with ultra-low particulate (ULPA) filter eliminate the contaminants in the air entering the clean room, it cannot control the internally-generated UFPs during the manufacturing processes (e.g., chemical vapor deposition (CVD), metallization, wet etching) (Choi et al., 2015; Manodori and Benedetti, 2009). If they are deposited on electrodes of a chip, they cause the interruption of the current flow, making the whole chip unusable and thereby reducing the yield of the semiconductor device (Libman et al., 2015). In this regard, ISO 14644-12 has been recently developed to guide how to monitor UFPs in cleanrooms. In order to monitor the concentration field of UFPs in these environments where the spatial and temporal variations of UFP concentrations are enormous, a portable and low-cost sensors are required to establish simultaneous monitoring at multiple points or establish dense monitoring networks.

**Question 5**

p. 1, l. 22: Some reference in the introduction seem for me to be very old, many 199Xs. There might be some very fundamental among those, but the last 20 years definitely brought some progress. Please check if there are newer references.

**Answer 5**

Thank you for your advice. We added recent references to the first paragraph of the introduction.

**We modified 'Introduction' part of the revised manuscript as following,**

Monitoring of airborne ultrafine particles (UFPs), which are smaller than 100 nm, is needed in various fields for human health and yield enhancement in industrial fields(Donaldson et al., 1998; Donovan et al., 1985; Hristozov and Malsch, 2009; Li et al., 2016; Liu et al., 2015). UFPs are mainly generated from burning fossil fuels and are ubiquitous in urban air; they account for about 90% of the total particle number concentration(Kim et al., 2011; Kittelson, 1998; Shi et al., 1999). Because of dramatic developments in nanotechnology, engineered UFPs for commercial and research purposes have been produced at a large scale. These incidentally and intentionally generated UFPs are more harmful to human health than larger counterparts: UFPs have a higher chance to deposit in the lower respiratory system and are more toxic owing to their larger surface-to-volume ratios, which causes oxidative stress, pulmonary inflammation, and tumor development(Hesterberg et al., 2012; Hext, 1994; Li et al., 2003; Renwick et al., 2004). Thus, onsite monitoring is needed to assess and minimize UFP exposure. High-precision industries with cleanrooms also need UFP monitoring to increase the production yield. For instance, in the semiconductor industry, the UFP minimum linewidth of the chips is approaching 7 nm(Neisser and Wurm, 2015). Particles that are a few nanometers in size are critical because "killer particles" (i.e., the diameter is greater than half of the minimum linewidth) can render the whole chip unusable(Libman et al., 2015). Unfortunately, since UFPs in cleanrooms are generated during fabrication processes (e.g., chemical vapor deposition (CVD), metallization, wet etching), contamination can occur in any manufacturing stages(Choi et al., 2015; Manodori and Benedetti, 2009). In these circumstances, a portable and low-cost sensor is needed for onsite UFP monitoring to accurately evaluate adverse health effects and control the contamination level in cleanrooms to enhance the production yield.

**We have added the references in the 1st paragraph of the introduction in the revised manuscript.**

(1) Li, N., Georas, S., Alexis, N., Fritz, P., Xia, T., Williams, M. A., Horner, E., and Nel, A.: A work group report on ultrafine particles (American Academy of Allergy, Asthma & Immunology): Why ambient ultrafine and engineered nanoparticles should receive special attention for possible adverse health outcomes in human subjects, J Allergy Clin Immunol, 138, 386-396, 2016.

(2) Liu, L., Urch, B., Poon, R., Szyszkowicz, M., Speck, M., Gold, D. R., Wheeler, A. J., Scott, J. A., Brook, J. R., Thorne, P. S., and Silverman, F. S.: Effects of ambient coarse, fine, and ultrafine particles and their biological constituents on systemic biomarkers: a controlled human exposure study, Environ Health Perspect, 123, 534-540, 2015.

(3) Kim, K. H., Sekiguchi, K., Kudo, S., and Sakamoto, K.: Characteristics of Atmospheric Elemental Carbon (Char and Soot) in Ultrafine and Fine Particles in a Roadside Environment, Japan, Aerosol and Air Quality Research, 11, 1-12, 2011.

(4) Hesterberg, T. W., Long, C. M., Bunn, W. B., Lapin, C. A., McClellan, R. O., and Valberg, P. A.: Health effects research and regulation of diesel exhaust: an historical overview focused on lung cancer risk, Inhal Toxicol, 24 Suppl 1, 1-45, 2012.

**Question 6**

p. 1, l. 23: "UFPs are mainly generated from burning fossil fuels …". This statement is not generally true, the main particle formation process is gas to particle formation and fossil fuel burning might be dominant in cities only.

**Answer 6**

We thank for your advice and letting us improve the solidity of the manuscript.

**We modified the 1st paragraph of 'Introduction' in the revised manuscript as following,**

Monitoring of airborne ultrafine particles (UFPs), which are smaller than 100 nm, is needed in various fields for human health and yield enhancement in industrial fields (Donaldson et al., 1998; Donovan et al., 1985; Hristozov and Malsch, 2009; Li et al., 2016; Liu et al., 2015). While they have a variety of anthropogenic and natural source, in urban area, UFPs are largely generated from the vehicle exhaust (e.g., soot agglomerates, secondary particles from hazardous gaseous precursors) (Kim et al., 2011; Kittelson, 1998; Shi et al., 1999). Moreover, because of dramatic developments in nanotechnology, engineered UFPs for commercial and research purposes have been produced at a large scale. These incidentally and intentionally generated UFPs are more harmful to human health than larger counterparts: UFPs have a higher chance to deposit in the lower respiratory system and are more toxic owing to their larger surface-to-volume ratios, which causes oxidative stress, pulmonary inflammation, and tumor development (Hesterberg et al., 2012; Hext, 1994; Li et al., 2003; Renwick et al., 2004). Thus, onsite monitoring is needed to assess and minimize UFP exposure.

**Question 7**

p. 2, l. 6: Please insert "e.g." before mentioning the TSI 3007". There are other models as well, e.g. from KANOMAX.

**Answer 7**

**We modified 'Description of the MEMS-based CPC' part of the revised manuscript as following,**

However, commercially available CPCs are bulky and expensive; thus, they are impractical for onsite monitoring where the UFP concentration changes continuously. Although portable CPCs (e.g. model 3007, TSI Inc., USA) are currently on the market, they are still large in size (292 mm × 140 mm × 140 mm) and expensive (~10,000 USD) for ownership. Therefore, despite their advantages, CPCs are difficult to actively utilize for onsite monitoring applications.

**We modified 'Conclusion' part of the revised manuscript as following,**

In terms of compactness and cost-efficiency, our system is superior to conventional instruments. MEMS-based CPC has a far smaller physical volume than the reference CPCs and is less than 91.5 % of the portable CPC (e.g. model 3007, TSI Inc., USA) available in the market.

**Question 8**

p. 2, l. 35: I might have missed it, but you don´t specify in the whole manuscript which working fluid was used. Did you try different ones? This information is essential for this technical paper and should be provided to the reader.

**Answer 8**

We thank for letting us catch the mistakes in our manuscript. The kind of working fluid should be clearly stated in manuscript. We used Butanol as the working fluid of MEMS based CPC.

**We modified '2 Description of the MEMS-based CPC' part of the revised manuscript as following,**

Figure 1 shows the operating principle of MEMS-based CPC, which consists of a reservoir, saturator, condenser, and miniature OPC. To generate supersaturated vapor and hence grow UFPs to micro-sized droplets, our system utilizes a conductive cooling method. Butanol was used as working fluid. The saturator generates saturated vapor by heating the wetted wall with the working fluid.

**Question 9**

p. 2, l. 40: In the supplement is only one figure. It seems strange to me to have a supplement because of just one figure. I suggest to incorporate it into the main manuscript.

If I remember correctly, in the traditional CPCs the particles grow more or less to the same droplet size, but in this new type there is a strong increase in droplet size with initial particle diameter (Fig. S1). How far does this go? Are even 20 or 30 µm droplets generated? How does this affect the counting efficiency of your new CPC for larger particles (sedimentation)? Please comment in the manuscript on that, if this is an issue.

p. 5, l. 24: Again, how large do the droplet in your CPC get at maximum and is sedimentation really no problem?

**Answer 9**

We thank for your careful comments. Following your advice,

**We moved the results of the droplet size distribution from the supplemental material to the revised manuscript, and have added the new section, '5.1 Droplet size distribution' in the result of the revised manuscript as following,**

Figure 6 shows the size distribution of the droplets generated from the MEMS-based condensation chip. Monodisperse Ag particles in the size range from 20 to 140 nm were used as test aerosol and their number concentrations were fixed at around 2000 cm$^{-3}$ by adjusting the valves of the dilution bridge. The sampling time for measuring each droplet distribution was 2 min, and the corresponding measurement uncertainty based on the Poisson statistics was 0.13 %. All the error bars at each data point represent the standard deviations. The commercial OPC (OPC-N2, Alphasense, UK) was used for measuring the droplet size distribution. It was reported that OPC-N2 was capable of not only measuring particles from 0.4 to 17.0 μm, but also having moderate counting performance compared to the reference OPC (PAS-1.108, Grimm Technologies) (Sousan et al., 2016). The measurement errors induced from the Mie resonance were not considered in this data. The average droplet diameter ($d_{d,avg}$) was 3.1 μm when particles with the size of 20 nm, slightly larger than the minimum detectable size (12.9 nm), were introduced. Since the lower detectable size of the optical detector in our system was 0.3 μm, introduced particles successfully grew into micrometer-sized droplets which were large enough to be counted by optical means. It was noted that the mean droplet size did not vary significantly above 40 nm. Also, most of the grown droplets were smaller than 10 μm, indicating that tens of micrometer-sized droplets, which could be attached to the inner walls of the condensation chip or optical detector via sedimentation, were barely generated.

[Figure]

Figure 6: The size distribution of the droplets grown from the MEMS-based condensation chip when Ag particles of specific sizes were introduced.

**We have added the reference which refers to the performance of OPC-N2 used in this study for measuring droplet size distribution in the revised manuscript.**

Sousan, S., Koehler, K., Hallett, L., and Peters, T. M.: Evaluation of the Alphasense Optical Particle Counter (OPC-N2) and the Grimm Portable Aerosol Spectrometer (PAS-1.108), Aerosol Sci Technol, 50, 1352-1365, 2016

**To evaluate the droplet loss in the condenser via sedimentation, we have performed the experiment, and added the result in '5.5 Performance comparison with the reference CPC' of the revised manuscript as followings,**

Figure 10 shows the measurement results of our system when it was tilted like an inset image. Monodisperse Ag particles with 25 nm were introduced and their concentrations were step-wisely increased from 0 to 4000 cm$^{-3}$. Since the measurement was carried out for about 500 s at each angle, and the measurement uncertainty of each section was below 0.01 %. When our system was oriented perpendicular to the surface, the counting efficiency of our system was 2.04 %. When a 30° angle was applied, the counting efficiency was 7.07 %. At 60°, the measurement difference compared to the

reference CPC exceeded 10 % (16.3 %). Thus, it was found that, at a tilt angle of 60º or less, MEMS-based CPC can monitor UFPs without the significant degradation of the accuracy.

The deviation of the counting efficiency induced from applying a tilt angle can be explained by the sedimentation of droplets in the condenser. At 0º, since the gravity direction was identical to the direction of the sample flow, the probability that grown droplets impacted on the condenser wall via sedimentation was negligible. However, with the increment of the tilt angle, the velocity vector of a droplet perpendicular to the channel increased, which lead to the decrement of the counting efficiency.

[Figure]

10    Figure 10: The time series of the number concentrations measured by MEMS-based CPC system when it was tilted.

**Question 10**

p. 3, l. 19: I did not fully understand what the micropillars in the condenser do. They are needed to prevent droplet formation, which could clog the channel, fine, but how exactly do they do this? Please explain more in detail.

15       p. 3, l. 22: Why is the pitch between the micropillars in the condenser not provided as number? All other dimensions are provided.

**Answer 10**

On rough surface, such as the surfaces with the micropillar array, wettability of liquid on the surface is increased.
20    Wenzel proposed an equation for the actual contact angle ($\theta_e$) as a function of static contact angle ($\theta_c$), which can be expressed as

$$\cos \theta_a = r cos\theta_s \ ,$$

where r is the roughness factor, the ratio of the actual area to the projected area of the surface. This equation indicates that the micropillar array decrease the effective contact angle, meaning that it increases the wettability of working fluid and thereby suppresses the droplet formation on the wall.

**We revised the content on the micropillar wick in '2 Description of the MEMS-based CPC' of the revised manuscript as followings,**

In the condenser, while supersaturated vapor grows UFPs to droplets, some may condense on the wall and clog the channel. Thus, like the saturator, the condenser also had micropillar-type wicks. On the rough surface, the actual contact angle of a working fluid droplet is lower than the contact angle on smooth surface (Chen et al., 2013). Thus, micropillar array increases the wettability of working fluid, suppressing the droplet formation on the wall and draining the condensed working fluid to the reservoir. While the diameter and length were the same, the pitch of the micropillar-type wick (130 μm) was larger than that in the saturator (100 μm).

**We added the geometric parameters of the MEMS-based condensation chip of the supplemental material as followings,**

[Figure]

Figure S1: The geometry parameters of the MEMS-based condensation chip.

**Question 11**

p. 4, l. 20: How long were the sampling lines? Which flow splitter was used? How is the flow geometry there? Did you have the same volume flow to all instruments (probably not, see Fig. 4)? A flow splitter can introduce strong deviations in the particle number concentration for different instruments connected to the flow splitter, in particular when using different volume flows, because the particles are not necessarily distributed homogenously over the sampling line. Please add the information on how the flow split exactly looks like and how you guaranteed that all three instruments good the same particle number concentration. According to Fig 4c I would guess for very small particles this was not the case.

p. 3, l. 29: If the Reynolds number in the system is so low (below 32) don´t you get problems with secondary flows, i.e. convection? Could you check this for instance using the Richardson number? How are the diffusional losses for such a flow? Please provide some numbers.

**Answer 11**

[Figure]

Figure R1: The optical image of the experimental setup for the performance test of our system

Figure R1 shows the optical image of the experimental setup. The CPC and aerosol electrometer were not used simultaneously; while aerosol electrometer was used for characterizing the counting efficiency and detectable

10    concentration range, CPC was used for the performance comparison.  Because of the large difference between the flow rates of the reference instrument and our system, the tube length of the reference instrument (either CPC or aerosol electrometer) was about 7 times longer than that of our system to guarantee the same transportation time. The following procedures were carried out to verify that particles with the same concentration were introduced into the two systems. First, to minimize the particle loss induced from the turbulence at the bifurcation, a flow splitter with a very small angle

15    of cleavage (model 3708, TSI Inc., USA) was used. The tubes which leads to the both systems were electrostatic dissipative to minimize the electrostatic particle loss, and their lengths were carefully adjusted to match the transportation times. To verify that the particles which introduced into both systems have the same concentrations, it was confirmed that the counting efficiency was close to 100 % when particles with size of 100 nm were introduced (it was assumed that they were activated and grew into droplets with 100 % efficiency). Then, while reducing the size of the introduced particles to

20    40 nm by adjusting the voltages of a DMA, it was confirmed whether the counting efficiency remained constant. Through these procedures, it was verified that the concentrations of the particles delivered to the two systems were the same.

Instead of characterizing the loss using an analytical method, the loss of our system was characterized using the counting efficiency, since it is defined as the efficiency of the system at detecting the introduced particles, and thereby

25    describes the overall transportation/activation efficiencies.

**We added the procedures and assumptions for characterizing the loss and counting efficiency in 'Experimental setup' part of the revised manuscript as following,**

Because of the large difference between the flow rates of the reference instrument and our system, the following procedures were carried out to verify that particles with the same concentration were introduced into the two systems. First, to minimize the particle loss induced from the turbulence at the bifurcation, a flow splitter with a very small angle of cleavage (model 3708, TSI Inc., USA) was used. The tubes which leads to the both systems were electrostatic dissipative to minimize the electrostatic particle loss, and their lengths were carefully adjusted to match the transportation times. To verify that the particles which introduced into both systems have the same concentrations, it was confirmed that the counting efficiency was close to 100 % when particles with size of 100 nm were introduced (it was assumed that they were activated and grew into droplets with 100 % efficiency). Then, while reducing the size of the introduced particles to 40 nm by adjusting the voltages of a DMA, it was confirmed whether the counting efficiency remained constant. Through these procedures, it was verified that the concentrations of the particles delivered to the two systems were the same. The loss of our system was characterized using the counting efficiency, since it is defined as the efficiency of the system at detecting the introduced particles, and thereby describes the overall transportation/activation efficiencies.

**We modified the 'Figure 4' in the revised manuscript as followings,**

[Figure]

**Figure 4: Schematic of the experimental setup for evaluating the performance of the MEMS-based CPC.**

**Question 12**

p. 5, l. 7: The activation efficiency is described as "the condensation chip at growing droplets" which I do not understand or feel to say something wrong. The activation efficiency is the fraction of particles being activated to droplets in the condensation chip.

p. 5, l. 9: Please add "in particular for small particles below ca. 30 nm" after "… on particle size," because the mentioned dependencies are mainly valid for this range.

p. 4, l. 37: The "dry out-region" (maybe better "dry out region"?), how were they identified? The red areas in Fig. 5 "show" them, but I see no difference in the photo inside the read areas and outside.

**Answer 12**

**We modified '2 Description of the MEMS-based CPC' of the revised manuscript as following,**

where $\eta_{trans}$, $\eta_{act}$, and $\eta_{OPC}$ are the efficiencies of a particle passing through our system, growing droplets at the condensation chip, and the OPC at counting droplets passing through its sensing volume, respectively. Because these three efficiencies are strongly dependent on the particle size, in particular for small particle below ca.30 nm the counting efficiency must be characterized as a function of the particle size.

**We modified the mistype in '5.1 Working fluid transmission and evaporation' of the revised manuscript as following,**

The dry-out region clearly did not form when the surface temperature was equal to the designed saturator temperature (40 °C) and even reached 70 °C. At 80 °C, the front of the working fluid started to recede, so a dry-out region formed.

[Figure]

Figure 5: (a) Schematic of the capillary rise experimental setup; (b) selected video frames from the rise of the working fluid using micropillar-type wick; (c) the dry-out region formation as the surface temperature increased.

**Question 13**

p. 5, l. 16: Fig, 6, how often was the counting efficiency curve measured? The day to day slightly different set-up can influence the curve, hence it should be measured at least three times, ideally on different days. How were the temperature settings and how does the counting efficiency change with different temperature settings?

**Answer 13**

Although we conducted the counting efficiency experiment five times, the data points shown was a single measurement to make it look clear. We added the additional data points in the previous experiments.

Counting efficiencies for various temperature differences between the saturator and condenser at 20, 25 and 30 ºC were characterized. The saturator temperature was not increased above 40 ºC, which was the maximum operating temperature of the miniature OPC.

**As your advice, we reinforced data in '5.3 Size-dependent particle counting efficiency' of the revised manuscript as following,**

[Figure]

Figure 7: Particle counting efficiency of the MEMS-based CPC as a function of the particle size for various saturator temperature. The particle size at which the particle counting efficiency was fitted to 50% was 12.9 nm (Ts = 40 ° C), 17.3 (Ts = 35 ° C) and 20.4 (Ts = 30 ° C), respectively.

Figure 7 shows the size-dependent counting efficiency of the MEMS-based CPC. The size range of Ag particles was controlled concentration range to 1000-2000 cm$^{-3}$. The sampling times for each data point were 300 s, and the measurement uncertainty based on the Poisson statistics was 0.02 %. To evaluate the effect of the temperature difference, the counting efficiency was characterized when the condenser temperature ($T_c$) was 10 ° C and the saturator temperatures ($T_s$) were 30, 35 and 40 ° C. At 40 ° C (the design value of the saturator temperature), the same experiments were repeated three times to confirm the measurement reliability. When the saturator temperature was 40 ° C, it was found that our system detected 1% of UFPs with the size of 5 nm, and the detection efficiency increased sharply above 9 nm. This was primarily because the activation efficiency ($\eta_{act}$) increased when the particle size exceeded the Kelvin diameter. The transport efficiency ($\eta_{trans}$) also increased, because the diffusivity of a particle decreases with the increment of the particle size. The counting efficiency data were curve-fitted using

$$\eta_d = \alpha + \frac{(\beta - \alpha)}{1 + (d_p/\gamma)^\delta} \, ,$$

where $\alpha$, $\beta$, $\gamma$ and $\delta$ are fitting constants of 101.96, 2.00, 12.99 and 4.70, respectively. The corresponding minimum detectable size is defined as the size at which particles are detected with 50 % efficiency and was found to be 12.9 nm. The detection efficiency was 90 % at 20.1 nm and reached 95% at 22.9 nm. It was close to 100 % and constant in the size range from 25 to 140 nm, indicating that the internal particle loss in this size range was negligible.

**Question 14**

p. 5, l. 20: "diffusivity of particles is inversely proportional to the size", is this true? How about the slip correction which brings a non-linear term into the particle diffusion problem?

**Answer 14**

We thank for letting us catch our mistake.

**We corrected the wrong sentence '5.3 Size-dependent particle counting efficiency' in the revised manuscript as following,**

The transport efficiency ($\eta_{trans}$) also increased, because the diffusivity of a particle decreases with the increment of the particle size.

**Question 15**

Fig. 1: I believe the miniature OPC scheme is incomplete. I cannot imagine that this OPC works in the forward scattering mode without using a beam blocker. Is this really true?

Fig. 5: Please add the unit "s" to the times provided above the photos.

Fig. 6.: Please provide uncertainty bars to the plot.

**Answer 15**

Figure R2 shows the (a) exploded view, (b) section A-A' and (c) section B-B' of the optical detector. It consists of the sensing chamber and optics (laser, cylindrical lens, elliptic mirror, optical detector, light trap). Introduced droplets are firstly arranged in a row in the acceleration nozzle (i.e., the outlet of the condensation chip) and enter the sensing chamber. The droplets then pass through the place where the condensed thin beam is irradiated. The mirror collects the scattered light from a droplet and redirect it to the optical detector.

When the laser beam passes through the cylindrical lens, the shape of the laser beam is not a point but a very thin surface. In addition, the acceleration nozzle at the chip outlet is only 0.8 mm in diameter and is located about 1.5 mm below the point where the beam passes. Therefore, as shown in Figure R2 (c), on the condition that the coincidence error does not occur (when two particles do not pass through the viewing volume at the same time), almost all the grown micro-droplets are counted in the optical detector.

In the perspectives of the structure and detection principle, the optical detector used in this study is similar to the high-precision OPC rather than dust sensors. Thus our system demonstrated particle counting performance, which was comparable to those of the reference CPC (model 3772, TSI Inc., USA).

[Figure]

Figure R2: The (a) exploded view, (b) section A-A' and (c) section B-B' of the optical detector used in this study

**We modified the schematic diagram of the miniature OPC in 'Figure 1' of the revised manuscript as following,**

[Figure]

Figure 1: Schematic illustration of the MEMS-based CPC. MEMS-based CPCr parts: the reservoir, saturator, condenser, and miniature OPC. The reservoir supplies the working fluid to the saturator via capillary action by the micropillar-type wick. The saturator heats the working fluid to generate saturated vapor. The saturated air becomes supersaturated when cooled by the condenser. UFPs grow into micro-sized droplets in the condenser and are counted by the miniature OPC.

**We modified the video frames in 'Figure 5' of the revised manuscript as following,**

[Figure]

Figure 5: (a) Schematic of the capillary rise experimental setup; (b) selected video frames from the rise of the working fluid using micropillar-type wick; (c) the dry-out region formation as the surface temperature increased.

Please refer to Answer 12. Error bars were added to the size-dependent particle counting efficiency graph.

**Question 16**

p. 1 1. 12: Please remove "the" before "3D".

p. 1, l. 17: Please replace "range of" with "range is", otherwise a verb would be missing.

p. 1, l. 17: The correct CF unit for the particle number concentration should be "1/cm3 ", without "N". Please correct in the whole manuscript.

p. 1, l. 22: A space is missing in before the "("…. This occurs several times in the manuscript, please correct.

p. 2, l. 7: Please remove "for ownership", this addition is not needed here.

p. 2, l. 18: Please remove the comma after "chip"

p. 2, l. 33: Please insert "to" before "grow".

p. 3, l. 2: Please exchange "proposed" with a different word, e.g. "new". The MEMS CPC is existing, you do not "propose" it.

p. 3, l. 7: Please insert "micropillar" before "dimensions".

p. 3, l. 18: Please insert "to" before "control".

p. 3, l. 19: "… some" what? "may condense on the wall", please specify that you mean the working fluid vapor.

p. 4, l. 14: Please exchange "They" with "The particles".

p. 5, l. 4: Please delete "drawn"

p. 5, l. 38: "the lower concentration limit of the aerosol electrometer was relatively high" Please rephrase, there is no "lower concentration limit" for an electrometer, however, because of the electronic noise you cannot trust the measured concentrations below a few hundred particles per cubic centimeter.

p. 7: The format of some references are different compared to the others, e.g., Hajjam et al, 2010 or Kim et al., 2015. There are more, please check all.

**Answer 16 - 1**

**We modified 'Abstract' part of the revised manuscript as following,**

A conventional conductive cooling–type CPC is miniaturized through MEMS technology and  3D printing technique, and the essential elements for growing droplets are integrated on a single glass slide.

**Answer 16 – 2**

**We modified 'Abstract' part of the revised manuscript as following,**

MEMS-based CPC measured the UFP number concentration with high accuracy (deviation within 4.1 %), and its detectable concentration range is 7.99–7200 cm$^{-3}$.

**Answer 16 – 3**

We will revise the relevant part of the manuscript according to the advice of the reviewer.

**Answer 16 – 4**

**We modified in the 1$^{st}$ paragraph of the introduction in the revised manuscript as following,**

Monitoring of airborne ultrafine particles (UFPs), which are smaller than 100 nm, is needed in various fields for human health and yield enhancement in industrial fields (Donaldson et al., 1998; Donovan et al., 1985; Hristozov and Malsch, 2009; Li et al., 2016; Liu et al., 2015).

**Answer 16 - 5**

**We modified 'Abstract' part of the revised manuscript as following,**

However, commercially available CPCs are bulky and expensive; thus, they are impractical for onsite monitoring where the UFP concentration changes continuously. Although portable CPCs (model 3007, TSI Inc., USA) are currently on the market, they are still large in size (292 mm × 140 mm × 140 mm) and expensive (~10,000 USD) .

**Answer 16 - 6**

**We modified in the 3$^{st}$ paragraph of the introduction in the revised manuscript as following,**

UFPs are grown to micrometer-sized droplets on the chip and the grown droplets are detected by the miniature OPC.

**Answer 16 - 7**

**We modified in the 3$^{st}$ paragraph of the introduction in the revised manuscript as following,**

To generate supersaturated vapor and hence to grow UFPs to micro-sized droplets,

**Answer 16 - 8**

We will revise the relevant part of the manuscript according to the advice of the reviewer.

**We modified 'Description of the MEMS-based CPC' part of the revised manuscript as following,**

By using the MEMS technology, our system can generate supersaturated vapor and grow droplets on a chip-scale system for significant decreases in the size, weight, and power consumption.

**Answer 16 - 9**

**We modified 'Description of the MEMS-based CPC' part of the revised manuscript as following,**

The dimensions of the micropillar-type wick were experimentally determined to be capable of pumping the working fluid from the reservoir and spreading it over the entire surface of the saturator to ensure that the saturator wall is always in the wetted condition.

**Answer 16 – 10**

Corresponding parts in the manuscript have been modified with the advice of previous reviewers as follows.

**We modified 'Description of the MEMS-based CPC' part of the revised manuscript as following,**

Figure 2a shows the customized circuit for our system. The circuit, whose dimension is 90 mm x 65 mm, simultaneously reads the data from the miniature OPC, temperature sensor and flow sensor (model 00H220H024, Nidec Co., JP), and controls the power of the heaters, cooling modules and micro pump (model FS1012-1020-NG, IDT Co., USA) via a pulse-width-modulation (PWM) method. In order for our system to be a stand-alone device, the feedback loops based on the proportional-integral-differential (PID) algorithm is implemented in the micro control unit (MCU) of the circuit, and their gains can be easily controlled using serial communication.

**Answer 16 – 11**

**We modified 'Description of the MEMS-based CPC' part of the revised manuscript as following,**

In the condenser, while supersaturated vapor grows UFPs to droplets, working fluid vapor may condense on the wall and clog the channel.

**Answer 16 – 12**

**We modified 'Experimental setup' part of the revised manuscript as following,**

The particles were electrically charged by a soft X-ray charger (XRC-05, HCT Co., KR) and then classified to a specific diameter with two types of DMA: (1) nano DMA (model 3085, TSI Co. Ltd., USA) for particles in the size range from 3 to 10 nm, (2) long DMA (model 3081A, TSI Co. Ltd., USA) for particles in the size range from 5 to 140 nm.

**Answer 16 – 13**

**We modified 'Experimental setup' part of the revised manuscript as following,**

The counting efficiency ($\eta_d$) is defined as the efficiency of the system at detecting the  particles and describes the overall CPC performance. It is the product of three efficiencies:

**Answer 16 – 14**

**We modified 'Detectable concentration range' part of the revised manuscript as following,**

As shown in Figure 7b, relatively large fluctuations were observed at number concentrations of $< 1000$ cm$^{-3}$ because of the electronic noise of the aerosol electrometer. However, at the number concentration range of 1000−5000 cm$^{-3}$, the overall difference between the concentrations of our system and the aerosol electrometer was only 4.1 %, which proves the high accuracy of our system.

**Answer 16 – 15**

We will revise the relevant part of the manuscript according to the advice of the reviewer.

**References**

Choi, K.-M., Kim, J.-H., Park, J.-H., Kim, K.-S., and Bae, G.-N.: Exposure characteristics of nanoparticles as process by-products for the semiconductor manufacturing industry, Journal of occupational and environmental hygiene, 12, D153-D160, 2015.

5  Donaldson, K., Li, X., and MacNee, W.: Ultrafine (nanometre) particle mediated lung injury, Journal of Aerosol Science, 29, 553-560, 1998.

Donovan, R., Locke, B., Osburn, C., and Caviness, A.: Ultrafine aerosol particles in semiconductor cleanrooms, Journal of The Electrochemical Society, 132, 2730-2738, 1985.

Hesterberg, T. W., Long, C. M., Bunn, W. B., Lapin, C. A., McClellan, R. O., and Valberg, P. A.: Health effects research and

10  regulation of diesel exhaust: an historical overview focused on lung cancer risk, Inhal Toxicol, 24 Suppl 1, 1-45, 2012.

Hext, P.: Current perspectives on particulate induced pulmonary tumours, Human & experimental toxicology, 13, 700-715, 1994.

Hristozov, D. and Malsch, I.: Hazards and risks of engineered nanoparticles for the environment and human health, Sustainability, 1, 1161-1194, 2009.

15  Kim, K. H., Sekiguchi, K., Kudo, S., and Sakamoto, K.: Characteristics of Atmospheric Elemental Carbon (Char and Soot) in Ultrafine and Fine Particles in a Roadside Environment, Japan, Aerosol and Air Quality Research, 11, 1-12, 2011.

Kittelson, D. B.: Engines and nanoparticles: a review, Journal of aerosol science, 29, 575-588, 1998.

Li, N., Georas, S., Alexis, N., Fritz, P., Xia, T., Williams, M. A., Horner, E., and Nel, A.: A work group report on ultrafine particles (American Academy of Allergy, Asthma & Immunology): Why ambient ultrafine and engineered nanoparticles

20  should receive special attention for possible adverse health outcomes in human subjects, J Allergy Clin Immunol, 138, 386-396, 2016.

Li, N., Sioutas, C., Cho, A., Schmitz, D., Misra, C., Sempf, J., Wang, M., Oberley, T., Froines, J., and Nel, A.: Ultrafine particulate pollutants induce oxidative stress and mitochondrial damage, Environmental health perspectives, 111, 455-460, 2003.

25  Liao, B.-X., Tseng, N.-C., Li, Z., Liu, Y., Chen, J.-K., and Tsai, C.-J.: Exposure assessment of process by-product nanoparticles released during the preventive maintenance of semiconductor fabrication facilities, Journal of Nanoparticle Research, 20, 2018.

Libman, S., Wilcox, D., and Zerfas, B.: Ultrapure Water for Advance Semiconductor Manufacturing: Challenges and Opportunities, ECS Transactions, 69, 17-28, 2015.

Liu, L., Urch, B., Poon, R., Szyszkowicz, M., Speck, M., Gold, D. R., Wheeler, A. J., Scott, J. A., Brook, J. R., Thorne, P. S.,

30  and Silverman, F. S.: Effects of ambient coarse, fine, and ultrafine particles and their biological constituents on systemic biomarkers: a controlled human exposure study, Environ Health Perspect, 123, 534-540, 2015.

Manodori, L. and Benedetti, A.: Nanoparticles monitoring in workplaces devoted to nanotechnologies, 2009, 012001.

Neisser, M. and Wurm, S.: ITRS lithography roadmap: 2015 challenges, Advanced Optical Technologies, 4, 235-240, 2015.

Renwick, L., Brown, D., Clouter, A., and Donaldson, K.: Increased inflammation and altered macrophage chemotactic

35  responses caused by two ultrafine particle types, Occupational and Environmental Medicine, 61, 442-447, 2004.

Shi, J. P., Khan, A., and Harrison, R. M.: Measurements of ultrafine particle concentration and size distribution in the urban atmosphere, Science of the Total Environment, 235, 51-64, 1999.

Sousan, S., Koehler, K., Hallett, L., and Peters, T. M.: Evaluation of the Alphasense Optical Particle Counter (OPC-N2) and the Grimm Portable Aerosol Spectrometer (PAS-1.108), Aerosol Sci Technol, 50, 1352-1365, 2016.

40  Stolzenburg, M. R. and McMurry, P. H.: An ultrafine aerosol condensation nucleus counter, Aerosol Science and Technology, 14, 48-65, 1991.

---

## Author Response (AR1)

**Microelectromechanical system-based condensation particle counter for real-time monitoring of airborne ultrafine particles**

Seong-Jae Yoo1, Hong-Beom Kwon1, Ui-Seon Hong1, Dong-Hyun Kang2, Sang-Myun Lee1, Jangseop Han1, Jungho Hwang1, Yong-Jun Kim1

[revised manuscript text omitted]

---

## Author Response (AR2)

[revised manuscript text omitted]

In figure (b): $N_{cal} = 47950 \times \ln N_n - 3649429$

6852 N cm⁻³ (Calibration start concentration) — 6852 N cm$^{-3}$ (Calibration start concentration)

| | Diameter (nm) | 28 | 26 | 24 | 22 | 20 | 18 | 16 |
|---|---|---|---|---|---|---|---|---|
| **Poisson statistics** | Time interval (s) | 96 | 66 | 84 | 72 | 108 | 108 | 66 |
| | Uncertainty (%) | 0.57 | 0.56 | 0.12 | 0.086 | 0.038 | 0.023 | 0.034 |
| **Number concentration (N cm⁻³)** | Reference CPC | 8.36 | 18.99 | 223.84 | 585.87 | 1370.69 | 3733.96 | 4129.93 |
| | MEMS-based CPC | 7.99 | 17.65 | 224.58 | 619.10 | 1399.14 | 3809.45 | 4544.82 |
| | Difference (%) | 4.42 | 7.60 | -0.32 | -5.37 | -2.03 | -1.98 | -9.12 |

[Figure]

**Figure 9: Time series of the number concentrations measured with our system and reference CPC when the concentration and size were varied.**

[Figure]

**Figure 10: Time series of the number concentrations measured by our system when it was tilted.**